



# Estimating vertically averaged energy dissipation rate

Nozomi Sugiura[1], Shinya Kouketsu[1], Shuhei Masuda[1], Satoshi Osafune[1], and Ichiro Yasuda[2]

[1]Research and Development Center for Global Change, JAMSTEC, Yokosuka, Japan
[2]Atmosphere and Ocean Research Institute, University of Tokyo, Chiba, Japan

*Correspondence to:* Nozomi Sugiura (nsugiura@jamstec.go.jp)

**Abstract.** The energy dissipation rate is an important characteristic of turbulence; however, its magnitude in observational profiles can be misidentified owing to its erratic evolution. By analysing observed data from oceanic turbulence, we show that the vertical sequences of depth-averaged energy dissipation rates have a scaling property, and propose a method to suitably estimate the vertically averaged value by utilizing that property. For scaling in the observed profiles, we found that averaging neighbouring points increases the expected value of its logarithm proportionally to the logarithm of the averaging interval. Furthermore, the population mean can be estimated for the logarithm of the vertically averaged energy dissipation rate from a single observation profile, by scaling up and promoting the observed value at each depth to one that corresponds to the whole profile. The estimate allows to distinguish whether an observational profile exhibits a momentarily high value by intermittency or maintains high energy dissipation on average.

## 1 Introduction

The clear importance of determining the energy dissipation rate to study ocean general circulation has been highlighted in many studies (e.g. Munk and Wunsch, 1998; Gregg et al., 1973). However, the calculation of its average magnitude in observational profiles can be easily misled by its erratic evolution. To determine the accurate state of turbulent energy, we need to clarify the definition of energy dissipation rate. To this end, we first define statistics regarding the energy dissipation rate, which is a term conveying different meanings including the local, depth-averaged, and ensemble-averaged rate, and hence should be properly established.

The local energy dissipation rate is defined as

$$\epsilon(\boldsymbol{x}, \omega) \equiv \frac{\nu}{2} \sum_{i,j=1,2,3} \left( \frac{\partial u_i}{\partial x_j} + \frac{\partial u_j}{\partial x_i} \right)^2, \tag{1}$$

where $\boldsymbol{u} = (u_1, u_2, u_3)$ is a velocity vector, $\nu$ is the kinematic viscosity, $\boldsymbol{x} = (x_1, x_2, x_3) = (x, y, z) = (\vec{x}, z) \in \mathbb{R}^3$ is a spatial coordinate, and $\omega$ represents a probabilistic event.

By taking the expectation, the ensemble-averaged energy dissipation rate is given by

$$\mathbb{E}[\epsilon(\boldsymbol{x})] = \int_0^\infty \epsilon(\boldsymbol{x}) p(\epsilon(\boldsymbol{x})) d\epsilon(\boldsymbol{x}), \tag{2}$$

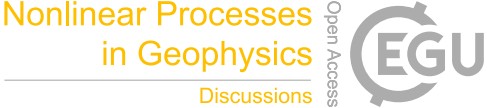



where $p(\epsilon(\boldsymbol{x}))$ is the probability distribution function of $\epsilon(\boldsymbol{x})$.

If we assign a value to a vertical segment $\left[z - \frac{r}{2}, z + \frac{r}{2}\right] \subset \mathbb{R}$ in an observed profile at horizontal location $\vec{x} \in \mathbb{R}^2$, the corresponding depth-averaged energy dissipation rate is given by

$$\epsilon_r(\boldsymbol{x}, \omega) = \frac{1}{r} \int\limits_{z' \in \left[z - \frac{r}{2}, z + \frac{r}{2}\right]} \epsilon(\vec{x}, z', \omega) dz'. \tag{3}$$

Energy dissipation rates obtained from oceanic observation (e.g. Itoh et al., 2009, and references therein) should be the depth-averaged ones (3) over some vertical segment with length scale $r$, because only the information from the segment is available, and points within the segment are indistinguishable.

Next, we discuss observational data averaging. In the study of ocean energetics, it is important to determine how turbulence energy dissipates on a large spatial scale and climate time scale, which should be represented by the 'average' energy dissipa-
tion rate (e.g., Waterhouse et al., 2014). Although previous studies have employed either arithmetic averaging, which conserves energy, or geometric averaging, which seems effective for a log-normal population (e.g., Whalen et al., 2015), no clear reasoning exists about the appropriate method of averaging observed energy dissipation rates, which would help us acquire a perspective on large-scale ocean mixing.

Geometric averaging of the vertically neighbouring values of $\epsilon_r$ in Eq. (3) gives

$$\widetilde{\epsilon_r}(\vec{x}_j, \omega') = \left( \prod_{k=1}^{K} \epsilon_r(\vec{x}_j, z_k, \omega_k) \right)^{1/K},$$
$$= \exp\left( \frac{1}{K} \sum_{k=1}^{K} \ln \epsilon_r(\vec{x}_j, z_k, \omega_k) \right), \tag{4}$$

where $z_k = r(k - \frac{1}{2})$ and $\omega' = \omega_1 \cap \omega_2 \cap \cdots \cap \omega_K$. Note that suffix $z$ on the left-hand side of Eq. (4) is dropped because it is defined for the profile. Assuming that the data in

$$\{\ln \epsilon_r(\vec{x}_j, z_k, \omega_k) \mid k = 1, 2, \cdots, K\} \tag{5}$$

approximately obey a Gaussian distribution for each $j$, $\widetilde{\epsilon_r}(\vec{x}_j, \omega')$ will serve as an estimate of the first parameter for the distribution of the energy dissipation rate at scale $r$ in a profile with length $\ell = Kr$.

In contrast, by arithmetically averaging the vertically neighbouring values of $\epsilon_r$ in Eq. (3), we could estimate the depth-averaged energy dissipation rate for a longer vertical segment:

$$\overline{\epsilon_\ell}(\vec{x}_j, \omega') = \frac{1}{K} \sum_{k=1}^{K} \epsilon_r(\vec{x}_j, z_k, \omega_k), \tag{6}$$

where $\ell = Kr$, $z_k = r(k - \frac{1}{2})$ and $\omega' = \omega_1 \cap \omega_2 \cap \cdots \cap \omega_K$. Observing that the $\omega_k$'s represent distinct events because we cannot simultaneously consider all $K$ observations at $z_k$'s, $\overline{\epsilon_\ell}(\vec{x}_j, \omega')$ corresponds to the depth-averaged energy dissipation rate of a coarse-grained event, which contains $K$ indistinguishable observations, instead of a single observational event.





Assuming that $\ln \epsilon_\ell(\vec{x})$ is populated according to a Gaussian distribution at a fixed horizontal location $\vec{x}$, we can estimate the population mean by averaging sample values $\ln \epsilon_\ell(\vec{x}, \omega_m)$, and the population mean is probably a key to the accurate estimation of energy balance. Hence, rather than the statistics given in Eq. (4) or (6), we should focus on the population mean for the logarithm of the vertical averages repeatedly sampled at each horizontal location. Specifically, assume that the data in

$\{\ln \epsilon_\ell(\vec{x}_j, \omega_m) \mid m = 1, 2, \cdots, M\}$   (7)

approximately obey a Gaussian distribution $\mathcal{N}(\mu_j, \sigma_j^2)$ for each $j$. The first parameter $\mu_j$ for the distribution of the energy dissipation rate at scale $\ell$ is estimated to be

$$
\begin{aligned}
\widetilde{\epsilon_\ell}(\vec{x}_j, \omega'') &= \left( \prod_{m=1}^{M} \epsilon_\ell(\vec{x}_j, \omega_m) \right)^{1/M}, \\
&= \exp \left( \frac{1}{M} \sum_{m=1}^{M} \ln \epsilon_\ell(\vec{x}_j, \omega_m) \right),
\end{aligned}
$$
(8)

which can be derived from a series of $M$ profile observations $\omega'' = \omega_1 \cap \omega_2 \cap \cdots \cap \omega_M$. However, we do not have access to this estimate because it requires many repeated observations at a fixed horizontal location, each one offering a vertically averaged instantaneous value $\epsilon_\ell(\vec{x}_j, \omega_m)$.

We propose an alternative for Eq. (8) to estimate the first parameter for the distribution of the vertically averaged energy dissipation rate for a profile by using a sequence of depth-averaged rates for shorter segments. To do so, we need to know the
statistical relation between the depth-averaged energy dissipation rates in scales $\ell$ and $r$, for $\ell \gg r$. To bridge different scales, we unravel the scaling law that underlies the observational data.

In this regard, we use some hints from previous studies. Kolmogorov (1962) claims that the velocity correlation of homogeneous isotropic turbulence is expressed as a function of the volume-averaged energy dissipation rate. Given the intermittency of turbulence, the logarithm of the depth-averaged energy dissipation rate obeys a Gaussian distribution, whose expected value
and variance depend linearly on the logarithm of the averaging scale. This claim suggests that the profiles of depth-averaged energy dissipation rates taken from any turbulence can have scale dependencies in general.

In this paper, we show that the observed data of depth-averaged energy dissipation rate have a scaling property and formalize a method to estimate the first parameter for the distribution of the vertical average by using this property. The remaining of the paper is organized as follows. In the next section, we analyse the scaling property and propose a calculation method for the
average. Then, we present calculation results by using real observational data. In the discussion, we describe the interpretation and importance of these results, and close the paper with conclusions on the importance of the proposed method and perspective for applications.

## 2   Proposed method

In this section, we describe the turbulence observation data employed in this study, formalize the scaling dependence encoded
in the data, and detail a method for calculating the vertical average accounting for the scaling relation.

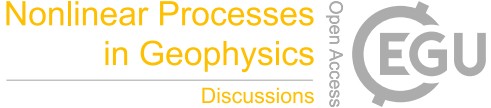

## 2.1 Observational data

The turbulence observational data were retrieved from the Pacific ocean (Fig. 1) (Goto et al., 2018) and comprise approximately $400$ profiles, each of which typically extends over $2000$ to $6000$ m-depth below the sea surface, with observational points every $5$ to $10$ m.

These positive-valued data have the following characteristics:

1. For sequential data in ordered set

$$\{\epsilon_r(\vec{x}_j, z_k) \mid k = 1, 2, \cdots, K\},$$

    each profile exhibits a very erratic evolution that impedes the recognition of a continuous curve along the depth direction (Fig. 2(a)).

2. After taking the logarithm of the values, the sequences seem more continuous (Fig. 2(b)). If we treat individually all the available data in nonordered set

$$\{\ln \epsilon_r(\vec{x}_j, z_k) \mid j = 1, 2, \cdots, J; k = 1, 2, \cdots, K\},$$

    the distribution of the logarithmic values approximately resembles a Gaussian distribution (Fig. 4), as pointed out by Gregg et al. (1973). We use this log-normality only in the approximate sense.

3. For the logarithmic sequence in ordered set

$$\{\ln \epsilon_r(\vec{x}_j, z_k) \mid k = 1, 2, \cdots, K\},$$

    after averaging each vertically neighbouring pair of values $\epsilon_r$ in a profile (Fig. 2(c)), the sequence shows larger values of $\ln \epsilon_{r'}$ than the original sequence on average:

$$\mathbb{E}_z\left[\ln \epsilon_{r'} - \ln \epsilon_r\right] > 0, \; r' = 2r,$$

where $\mathbb{E}_z$ represents averaging in any vertical interval larger than $r'$ (Fig. 2(d)).

4. If we repeat procedure 3, we obtain the sequences that have even larger values of $\ln \epsilon_{r'}$ on average (Fig. 3).

The above shift is a direct consequence of the inequality of arithmetic and geometric means, i.e.

$$\epsilon_{2r} = \frac{\epsilon_r^{(U)} + \epsilon_r^{(L)}}{2}$$
$$\geq \left(\epsilon_r^{(U)} \epsilon_r^{(L)}\right)^{1/2} = \exp\left(\frac{\ln \epsilon_r^{(U)} + \ln \epsilon_r^{(L)}}{2}\right),$$

for the upper, $(U)$, and lower, $(L)$, parts of $\epsilon_{2r}$. However, the shift amount can be a specific property of the analysed data. Furthermore, although the shifts $\ln \epsilon_{2r'} - \ln \epsilon_{r'}$ are fluctuating from point to point, it is suggested that the vertical average of the shifts for a profile does not differ much between all the observed profiles.

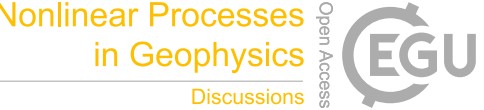



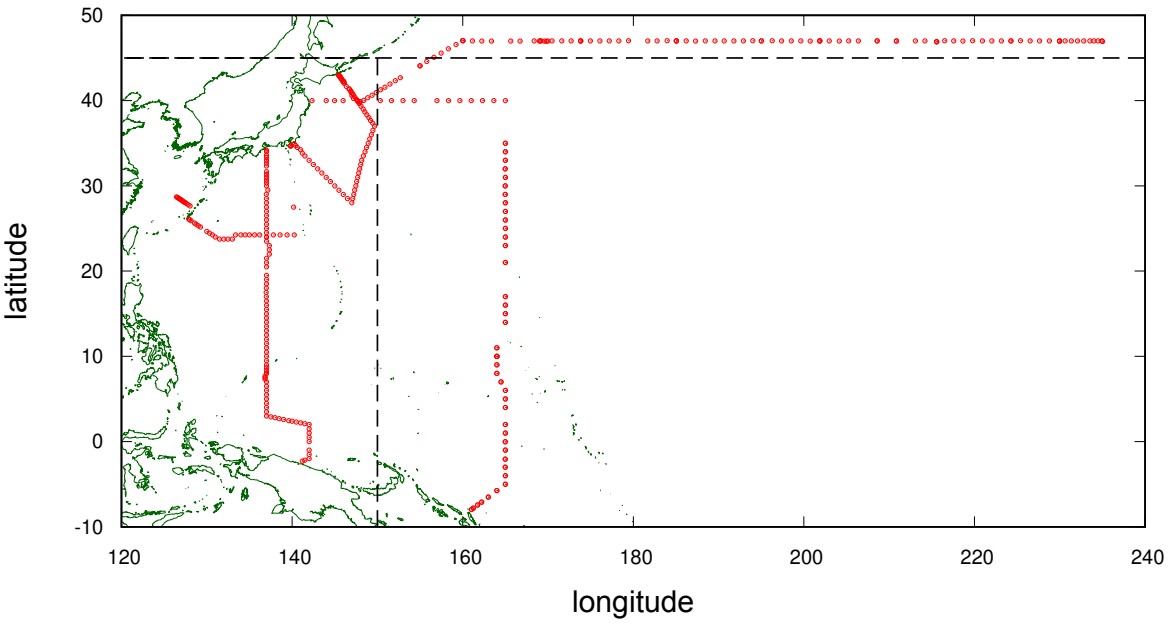

**Figure 1.** Horizontal locations of the observed profiles (red), and land-sea boundaries (green). The units of longitude and latitude are °E and °N, respectively.

## 2.2 Scaling analysis

The statistical phenomenon observed in Fig. 3 can be attributed to the scaling properties encoded in the data sequence from the profiles. Here, we introduce a random variable $\alpha$, which serves as the factor of proportionality, and postulate that averaging neighbouring points increases its logarithmic value proportionally to the logarithm of the averaging interval:

$$\ln \frac{\epsilon_{r'}(\overrightarrow{x}, z')}{\epsilon_r(\overrightarrow{x}, z)} = \alpha \ln \frac{r'}{r}, \tag{9}$$

where the right-hand side is $\alpha n \ln 2$ for $r' = 2^n r$. By taking the expectation, we obtain

$$\mathbb{E}\left[\ln \frac{\epsilon_{r'}(\overrightarrow{x}, z')}{\epsilon_r(\overrightarrow{x}, z)}\right] = \mathbb{E}[\alpha] \ln \frac{r'}{r}. \tag{10}$$

The target population here is all possible states of profiles at any horizontal points of observation. Thus, if we interpret $\mathbb{E}$ as averaging $\mathbb{E}_{\boldsymbol{x}}$ among all the available pairs of segments with length $r$ and $r'$ taken from observed profiles at various horizontal





(a) Original profiles $\epsilon_r$ in linear scale

(b) Original profiles $\ln \epsilon_r$

(c) Depth averages among every pair of neighbouring segments $\ln \epsilon_{r'}$, $r' = 2r$

(d) Difference $\ln \epsilon_{r'} - \ln \epsilon_r$

**Figure 2.** Appearances of observed profiles.

locations, then Eq. (10) serves as an estimator for the factor:

$$\widehat{\alpha} = \mathbb{E}_{\boldsymbol{x}} \left[ \ln \frac{\epsilon_{r'}(\vec{x}, z')}{\epsilon_r(\vec{x}, z)} \right] / \left( \ln \frac{r'}{r} \right). \tag{11}$$

Note that the estimated value $\widehat{\alpha}$ is accompanied by an estimation error, which will be discussed later in section 3.1.2.

If this assumption holds and factor $\widehat{\alpha}$ is calculated from observational data, then we obtain an estimator for the vertically

5   averaged energy dissipation rate in a larger segment:

$$\ln \widehat{\epsilon_{r'}}(\vec{x}, z') = \ln \epsilon_r(\vec{x}, z) + \widehat{\alpha} \ln \frac{r'}{r}, \tag{12}$$

which corresponds to several $\ln \epsilon_r(\vec{x}, z)$'s centred at $z$ satisfying $[z - r/2, z + r/2] \subset [z' - r'/2, z' + r'/2]$, and thus there are $2^n$ estimates per $\ln \epsilon_{r'}(\vec{x}, z')$. Estimated values $\widehat{\epsilon_{r'}}(\vec{x}_j, z_k)$ and the arithmetic mean $\overline{\epsilon_{r'}}(\vec{x}_j, z_k)$ calculated from observational



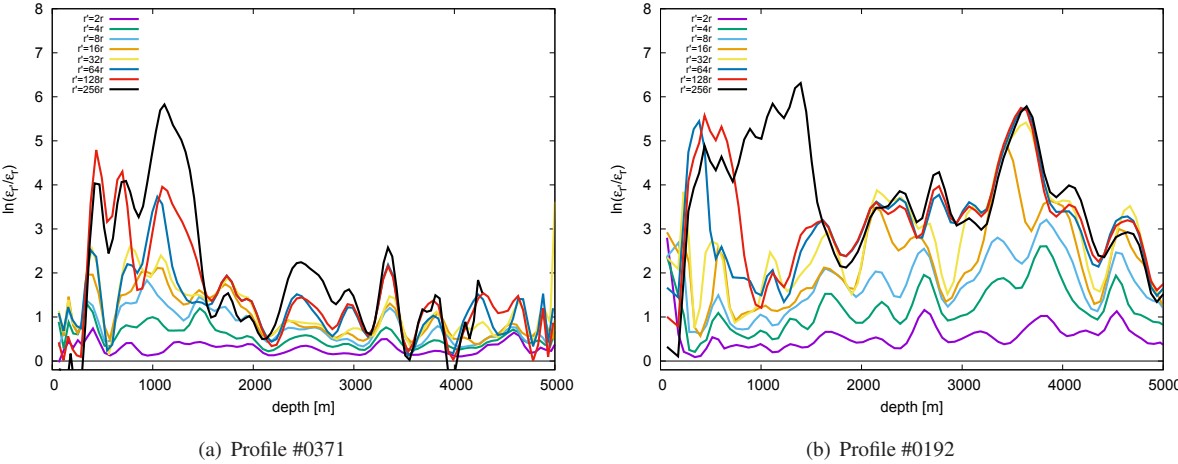

(a) Profile #0371    (b) Profile #0192

**Figure 3.** Evolution of averages $\mathbb{E}_z\left[\ln \epsilon_{r'} - \ln \epsilon_r\right]$ as $r'$ increases. For simplicity of presentation, Bèzier curves are used for average $\mathbb{E}_z$.

data satisfy

$$\prod_{j,k} \widehat{\epsilon_{r'}}(\vec{x}_j, z_k) = \prod_{j,k} \overline{\epsilon_{r'}}(\vec{x}_j, z_k), \tag{13}$$

where the geometric summation is taken for every pair of segments with lengths $r$ and $r'$ taken from profiles at various horizontal locations.

5    If the observational points in a profile are arranged with uneven spacing and their coordinates are approximated by a power function $z_k = ak^c + b,\ k = 0,\ 1,\ 2,\ \cdots$, the left-hand side of Eq. (9) should be corrected as $c^{-1}\ln\left(\epsilon_{r'}/\epsilon_r\right)$ for the right-hand side to share the common form $\widehat{\alpha}n\ln 2$. In detail, the scaling analysis based on Eq. (9) is performed as follows:

1. Suppose we have an observational sequence in ordered set

$$\{\epsilon_r(\vec{x}_j, z_k) \mid k = 1, 2, \cdots, K_j\}$$

   for each $j$.

2. By averaging vertically neighbouring $2^n$ points, we obtain ordered set

$$\{\epsilon_{r'}(\vec{x}_j, z_{k'}) \mid k' = 1,\ 2,\ \cdots, [K_j/2^n]\},$$

   where $r' = 2^n r$ and

$$\epsilon_{r'}(\vec{x}_j, z_{k'}) = 2^{-n} \sum_{k=2^n(k'-1)+1}^{2^n k'} \epsilon_r(\vec{x}_j, z_k).$$

   As a typical observational segment is $r \simeq 5$ m, with profile length 5000 m, $n$ is not larger than $10 \doteqdot \ln(5000/5)/\ln 2$.



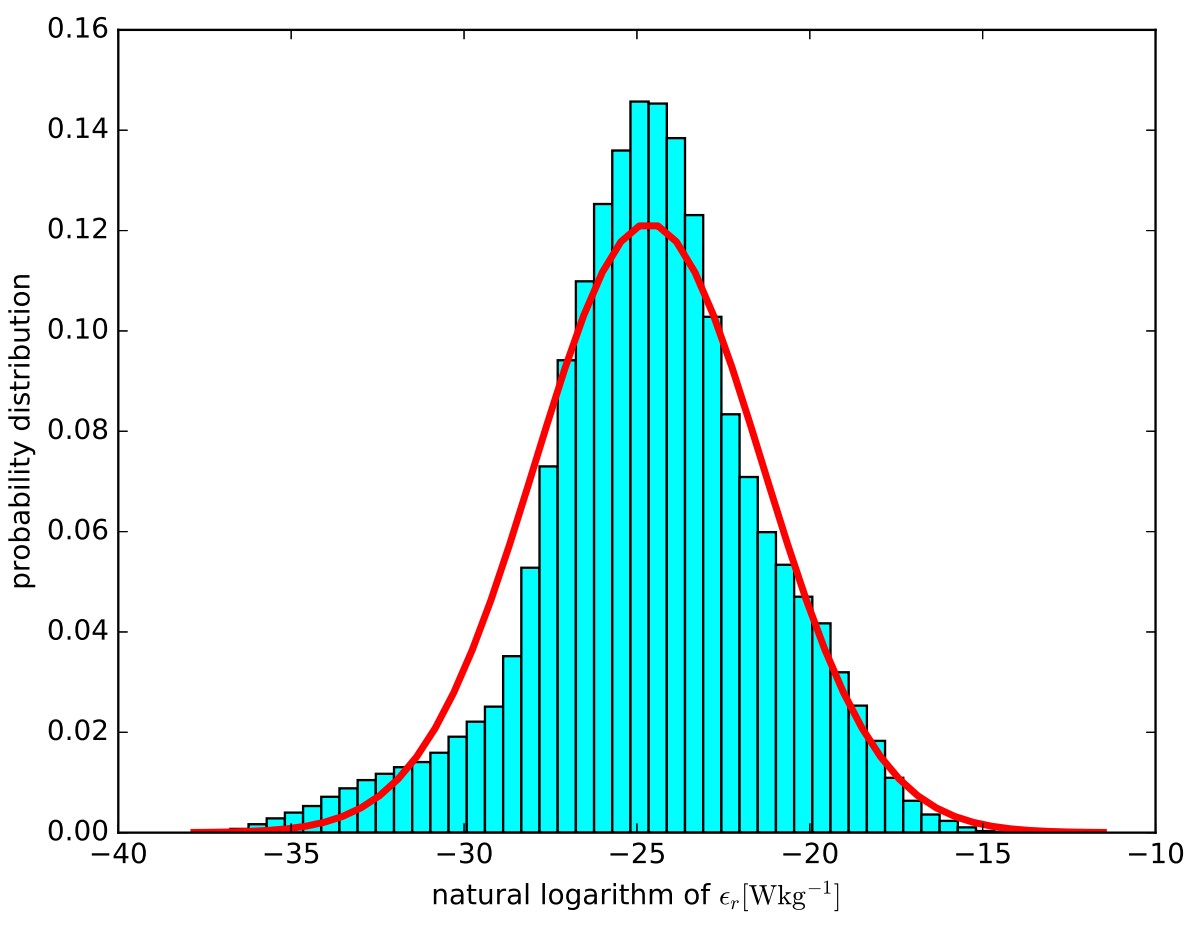

**Figure 4.** Distribution of logarithm of observational data (bar chart), and the best-fitting Gaussian distribution (red).

3. With respect to all possible triplets $(j, k, k')$ for fixed $n$, we calculate the average of

$$c_j^{-1} \left[ \ln \epsilon_{r'}(\vec{x}_j, z_{k'}) - \ln \epsilon_r(\vec{x}_j, z_k) \right],$$

where $c_j$ is the correction for the uneven observational spacing, and denote the average as $Y_n$.

4. Plot $(n \ln 2, Y_n)$, $n = 0, 1, 2, \cdots, 10$ and find the best-fitting slope. Considering measurement error, the fitting is performed as a curve (see Appendix A).

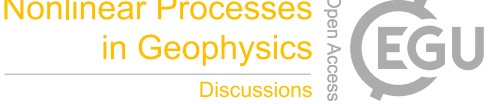

### 2.3 Rescaled average

Suppose $S(\vec{x})$ represents the observation of a profile at coordinate $\vec{x}$, i.e. the set of all pairs of segments with length $\ell$ and $r$ in a profile of length $\ell = Kr$. Assuming statistical homogeneity of $\alpha$ in Eq. (9) and applying it to the events in $S(\vec{x})$, the conditional expectation is

$$5 \quad \mathbb{E}_{\boldsymbol{x}}\left[\ln \frac{\epsilon_\ell}{\epsilon_r}\bigg| S(\vec{x})\right] = \widehat{\alpha}\ln \frac{\ell}{r} = \widehat{\alpha}\ln K. \tag{14}$$

This implies an estimator for the first parameter for the distribution of vertical average $\epsilon_\ell(\vec{x})$:

$$\widehat{\epsilon}_\ell(\vec{x}) \equiv \exp\left(\mathbb{E}_{\boldsymbol{x}}\left[\ln \epsilon_\ell | S(\vec{x})\right]\right)$$
$$= \exp\left(\frac{1}{K}\sum_{k=1}^{K}\ln \widehat{\epsilon}_\ell(\vec{x}, \omega_k)\right)$$
$$= \left(\prod_{k=1}^{K}\left(K^{\widehat{\alpha}}\epsilon_r(\vec{x}, z_k, \omega_k)\right)\right)^{1/K}, \tag{15}$$

where we used $\ln \widehat{\epsilon}_\ell(\vec{x}, z') = \ln \epsilon_r(\vec{x}, z) + \widehat{\alpha}\ln K$ for the third equality. Here, 'the first parameter' is for the log-normal distribution of the states that can occur during the profile observation. Note that in general, estimate $\widehat{\epsilon}_\ell(\vec{x})$ is neither equal to geometric mean $\widetilde{\epsilon}_r(\vec{x})$ in Eq. (4) nor to arithmetic mean $\overline{\epsilon}_\ell(\vec{x})$ in Eq. (6). Instead, it is similar to $\widetilde{\epsilon}_\ell(\vec{x})$ in Eq. (8) except that every $\ln \epsilon_\ell(\vec{x}, \omega_m)$ is substituted by the estimated $\ln \widehat{\epsilon}_\ell(\vec{x}, \omega_k)$, which resemble $K$ repeated observations (see Fig. 5). Hereafter, the estimate given by Eq. (15) is called the rescaled average.

We cannot necessarily assume statistical homogeneity of $\alpha$ along the horizontal or vertical direction. With this in mind, we also consider the conditional expectations using three different $\widehat{\alpha}$ values:

$$\mathbb{E}_{\boldsymbol{x}}\left[\ln \frac{\epsilon_\ell}{\epsilon_r}\bigg| S_i(\vec{x})\right] = \widehat{\alpha}_i\ln \frac{\ell}{r}, \tag{16}$$

where $i$ denotes either three horizontal regions, as those divided by dashed lines in Fig. 1, or two vertical regions above and below 200 m depth. Consequently, $\widehat{\alpha}_i$ is derived from the scaling analysis using the data from region $i$, and thus we have other
approximation formulae for $\epsilon_\ell(\vec{x})$:

$$\widehat{\epsilon}_\ell(\vec{x}) \equiv \exp\left(\mathbb{E}_{\boldsymbol{x}}\left[\ln \epsilon_\ell | S_i(\vec{x})\right]\right)$$
$$= \left(\prod_{k=1}^{K}\left(K^{\widehat{\alpha}_i}\epsilon_r(\vec{x}, z_k, \omega_k)\right)\right)^{1/K}, \tag{17}$$

according to the region that $\vec{x}$ belongs to.



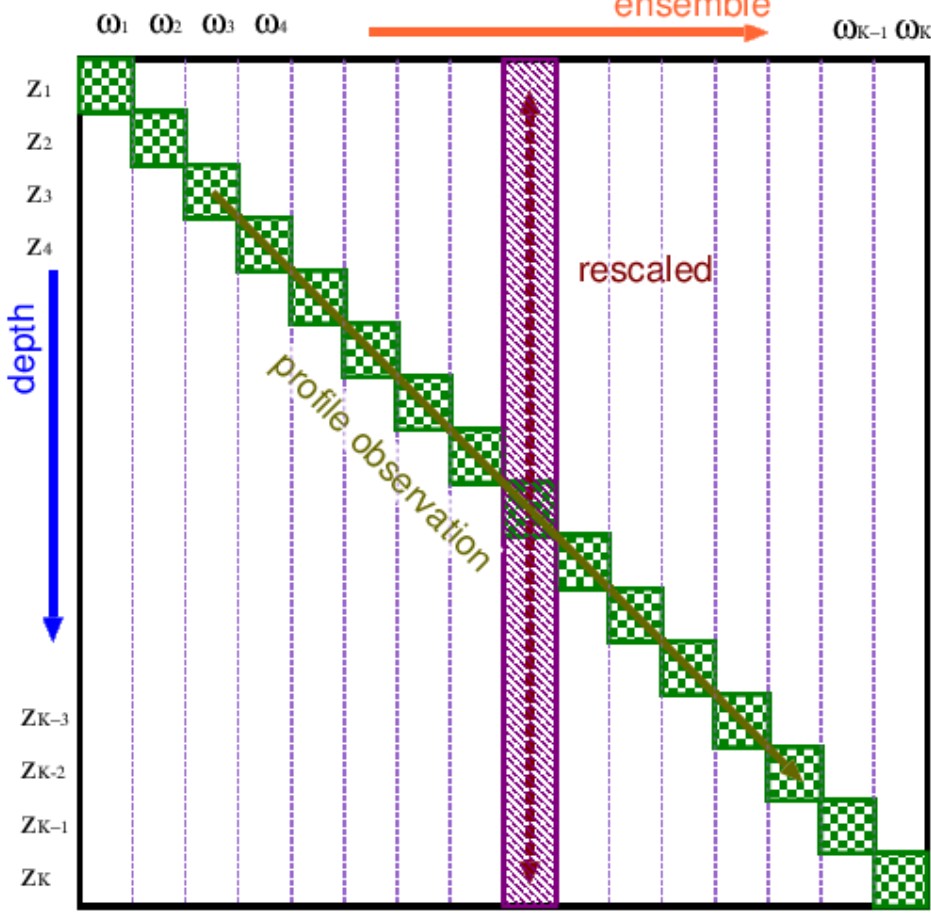

**Figure 5.** Concept of rescaled average. A single observation (green square) is rescaled to the profile length (purple rectangle) and treated as an ensemble member.

## 3 Results

### 3.1 Scaling analysis

#### 3.1.1 Estimated value

By using the observational data, we conducted the scaling analysis according to the procedure described at the end of section 2.2. The values of $\mathbb{E}_{\boldsymbol{x}}\left[\ln\left(\epsilon_{r'}/\epsilon_r\right)\right]$ are plotted against $\ln\left(r'/r\right)$ as black dots in Fig. 6, where a scaling relation with exponent $\widehat{\alpha} = 0.467$ valid from $r = 5$ m to $r = 5000$ m can be observed.

The comparison among data above and below 200 m shows that the scaling appears to be statistically homogeneous along the depth direction (top graph of Fig. 6). On the other hand, the comparison among data from the three horizontal areas does





not necessarily show that the scaling is homogeneous along the horizontal direction (bottom graph of Fig. 6). The estimated scaling exponents are $\widehat{\alpha}_1 = 0.375$ for the area with $y \geq 45\,^\circ$N, $\widehat{\alpha}_2 = 0.462$ for $y < 45\,^\circ$N and $x < 150\,^\circ$E, and $\widehat{\alpha}_3 = 0.530$ for $y < 45\,^\circ$N and $x \geq 150\,^\circ$E, where $x$ is the longitude and $y$ is the latitude.

These results confirm the claim that averaging neighbouring points increases the expected value of its logarithm by a rate, $\widehat{\alpha} > 0$ or $\widehat{\alpha}_i > 0$, proportional to the logarithm of the averaging interval. Moreover, we determined the values of the scaling exponent through the statistical analysis of the observational data.

### 3.1.2 Estimation error

For simplicity, here we only consider the case with a single $\widehat{\alpha}$. The scaling relation (11) provides a mean estimate for $\alpha$ from the data as

$$\widehat{\alpha} = \mathbb{E}_{\boldsymbol{x}}\left[A\right],\tag{18}$$

$$A \equiv \ln\left(\frac{\epsilon_{r'}(\overrightarrow{x}, z_{k'}, \omega_{k'})}{\epsilon_{r}(\overrightarrow{x}, z_{k}, \omega_{k})}\right)\bigg/\ln\frac{r'}{r}.\tag{19}$$

We can also calculate the variance of the estimated $\alpha$ from the data as

$$\sigma_\alpha^2 = \mathbb{E}_{\boldsymbol{x}}\left[A^2\right] - \mathbb{E}_{\boldsymbol{x}}\left[A\right]^2$$
$$= \mathbb{E}_{\overrightarrow{x}}\left[\mathbb{E}_{z}\left[A^2\right] - \mathbb{E}_{z}\left[A\right]^2\right] + \mathbb{E}_{\overrightarrow{x}}\left[\mathbb{E}_{z}\left[A\right]^2\right] - \mathbb{E}_{\overrightarrow{x}}\left[\mathbb{E}_{z}\left[A\right]\right]^2$$

$$\equiv \sigma_{\text{intra}}^2 + \sigma_{\text{inter}}^2,\tag{20}$$

where $\sigma_{\text{intra}}^2$ is the average of variances within a profile, and $\sigma_{\text{inter}}^2$ is the variance of profile-averages. Figure 7 shows that $\sigma_{\text{intra}}$ and $\sigma_{\text{inter}}$ are approximately $0.44$ and $0.20$, respectively, when $r' \gg r$.

## 3.2 Rescaled average

### 3.2.1 Estimated value

The vertical average of the energy dissipation rate for the upper $\ell = 2000\,\text{m}$ at each horizontal location was calculated according to Eq. (15) or (17). The value of $\ell$ was selected such that the length of the calculated segments agrees with that of a typical observed profile. We excluded the profiles with less than 50 data points in the upper $2000\,\text{m}$.

In Fig. 8, the simple arithmetic average from Eq. (6) is compared with the rescaled estimates using a single $\alpha$ value (top graph) and three $\alpha$ values (bottom graph). In most points, the rescaled estimate shows larger values than the arithmetic average,

which still exhibits very large values intermittently, unlike the rescaled estimate. There seems to be no significant difference between the distributions in the top and bottom graphs of Fig. 8. These results suggest that we can suitably estimate the order of the values even using a single $\alpha$ value.



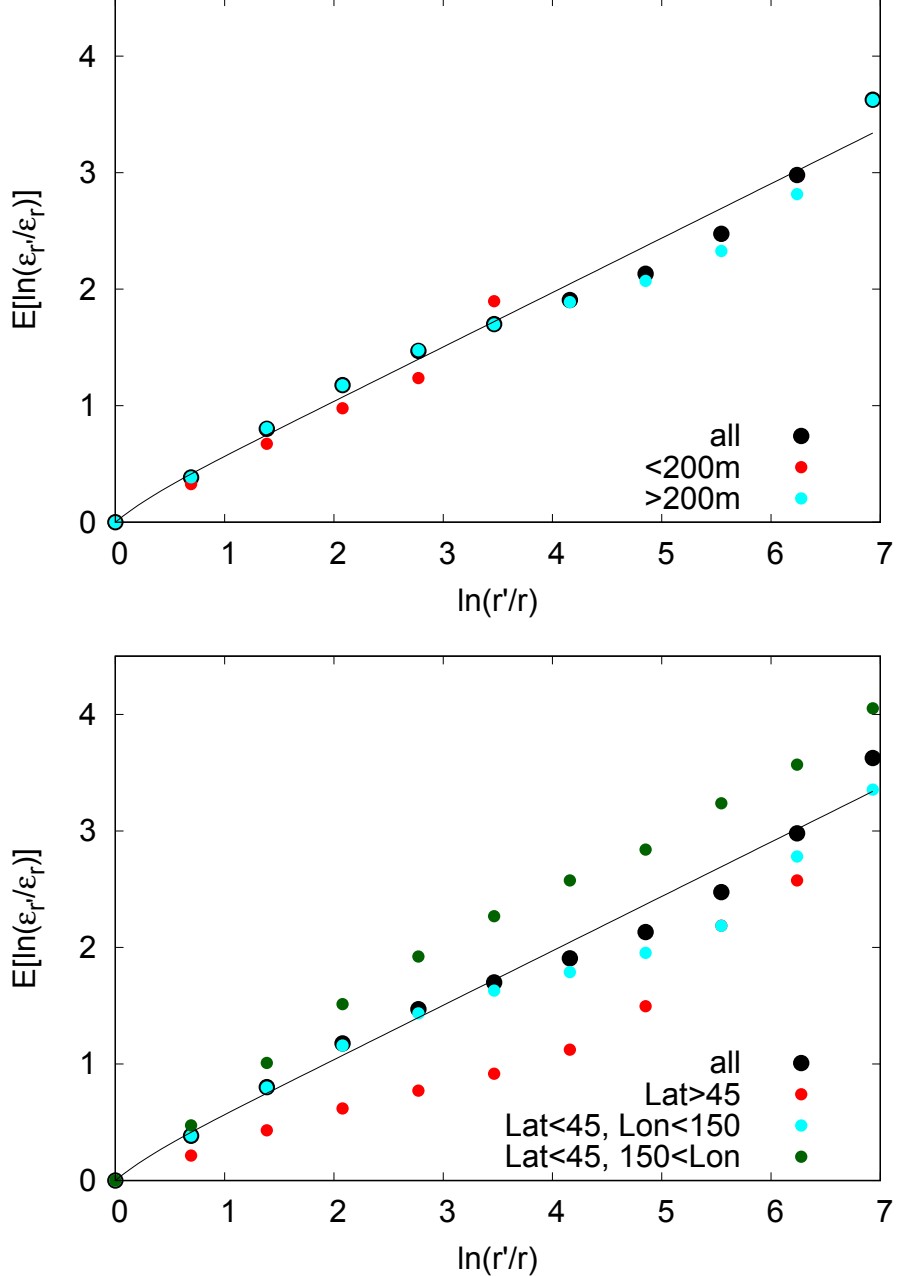

**Figure 6.** Scaling property of observational data. Average of $\ln(\epsilon_{r'}/\epsilon_r)$ (dots) and best-fitting curve of the form $y = ax + b(1 - e^{-cx})$ using all the data points (solid line). The results were obtained for different depth sections (top graph), and horizontal sections (bottom graph).

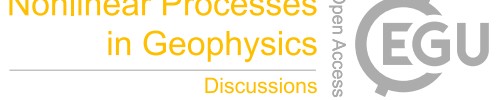



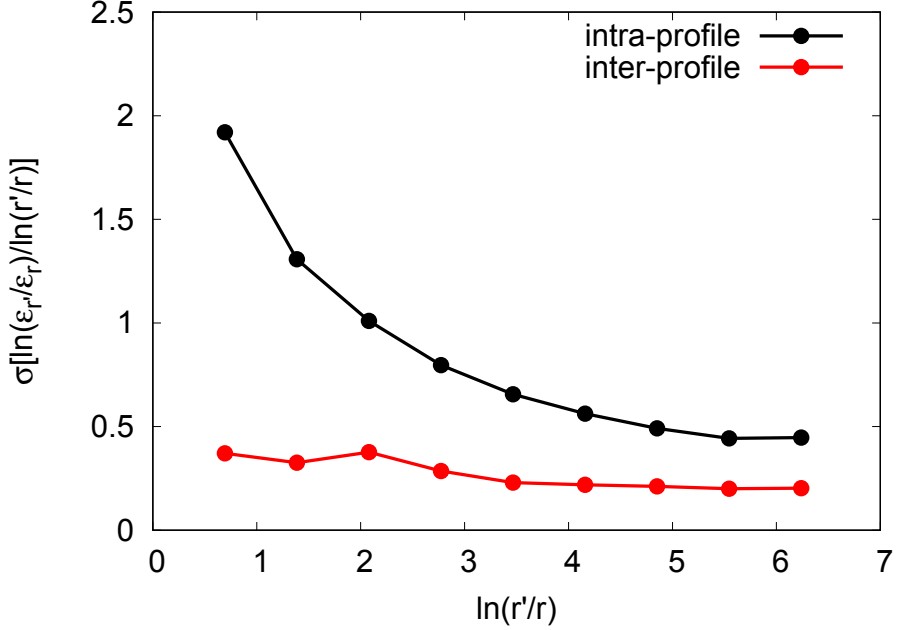

**Figure 7.** Estimation errors of $\widehat{\alpha}$ according to scales $\ln\left(r'/r\right)$. Intra-profile $\sigma_{\text{intra}}^2$ is the average of variances within a profile, and inter-profile $\sigma_{\text{inter}}^2$ is the variance of profile-averages.

### 3.2.2 Estimation error

Here, we only consider the case with a single $\alpha$ for simplicity, and fix a horizontal location $\vec{x}$. We assume that $\ln\epsilon_\ell(\vec{x})$ is populated according to an almost Gaussian distribution with unknown mean $\mu$ and variance $\sigma^2$, which are to be estimated.

Using the arithmetic averaging strategy, we obtain a single sample $\ln\overline{\epsilon_\ell} = \ln\left(K^{-1}\sum_{k=1}^{K}\epsilon_r(\vec{x},z_k)\right)$ with variance $\sigma_{\ln\overline{\epsilon_\ell}}^2 = \sigma^2$, and the likelihood of population mean $\mu$ is proportional to $\exp\left((\mu - \ln\overline{\epsilon_\ell})^2/(2\sigma_{\ln\overline{\epsilon_\ell}}^2)\right)$. Note that, basically, we cannot obtain any information about the estimation error.

By contrast, using the rescaled averaging strategy, we retrieve $K$ samples $y_k = \ln\epsilon_r(\vec{x},z_k)$, $k=1,2,\cdots,K$ from an almost Gaussian population with variance $\sigma_{\ln\epsilon_r}^2$, and rescale them into $y'_k = \ln\epsilon_r(\vec{x},z_k) + \alpha(\vec{x},z_k)\ln K$ at the cost of additional uncertainties of $\alpha$ for intra-profile and inter-profile. Hence, the likelihood of the population mean $\mu$ is proportional to





$\exp\left((\mu - \ln \widehat{\epsilon}_\ell)^2/(2\sigma^2_{\ln \widehat{\epsilon}_\ell})\right)$, where

$$\ln \widehat{\epsilon}_\ell = K^{-1} \sum_{k=1}^{K} y'_k$$

$$= K^{-1} \sum_{k=1}^{K} \ln \epsilon_r(\vec{x}, z_k) + \widehat{\alpha} \ln K, \tag{21}$$

$$\sigma^2_{\ln \widehat{\epsilon}_\ell} \simeq K^{-1}\sigma^2_{\ln \epsilon_r} + K^{-1}(\sigma_{\text{intra}} \ln K)^2 + (\sigma_{\text{inter}} \ln K)^2. \tag{22}$$

Inserting typical values in the observed data for $K \gg 1$, we obtain

$$\sigma^2_{\ln \widehat{\epsilon}_\ell} \simeq 2.6^2/K + (0.44 \ln K)^2/K + (0.20 \ln K)^2, \tag{23}$$

where $\sigma^2_{\ln \epsilon_r} = 2.6^2$ is the average of intra-profile variance of $\ln \epsilon_r$ calculated from observational data, and $\sigma_{\text{intra}}$ and $\sigma_{\text{inter}}$ are from sect 3.1.2. In particular, for $K = 2000\,\text{m}/5\,\text{m} = 400$, we have $\sigma_{\ln \widehat{\epsilon}_\ell} = 1.2$, which is $0.52$ in common logarithm.

Comparing the estimation errors $\sigma_{\ln \widehat{\epsilon}_\ell}$ and $\sigma_{\ln \overline{\epsilon}_\ell}$, although the rescaled average is not necessarily more accurate than the
arithmetic average, the former has explicitly defined uncertainty that remains relatively unchanged from profile to profile, which is an advantage as an estimate for the first parameter.

### 3.2.3  Horizontal distribution

Figures 9 and 11 show the horizontal distribution of the vertically averaged energy dissipation rate by using a single and three $\widehat{\alpha}$ values, respectively. A cross section along $47\,°\text{N}$ is shown in Fig. 10. The distributions of the areas with large $\epsilon_\ell$ value are
different in many regions between the rescaled and arithmetic averages, despite some similarities. High peaks are sometimes isolated for the arithmetic average, whereas they tend to form clusters for the rescaled average (see Fig.10), which has at most the order of $3 \times 10^{-8}$ to $1 \times 10^{-7}\ \text{W} \cdot \text{kg}^{-1}$ (see Figs. 9 and 11), suggesting that the rescaled average captures more robust features from the distribution of energy dissipation.

These results indicate that the calculated values from rescaled averaging can be used to correctly interpret the horizontal
distribution of energy dissipation.

## 4    Discussion

1. Previous studies have not clarified the relation between small-scale observation of turbulence in several meters with data substantially intermittent and the large-scale features of energy dissipation, which should be more persistent. In this paper, we propose a method to relate them quantitatively by elucidating the scaling relations underlying the observational
data and thereby estimating the vertical average through rescaled averaging.

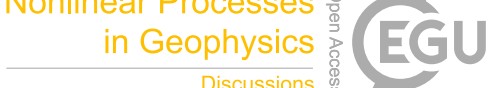

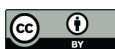

2. The volume-averaged energy dissipation rate can be considered as the value when the local energy dissipation rate, a Schwartz distribution, is evaluated against a test function $\phi$:

$$\epsilon(\phi_{z_0}^{\lambda}) = \int \epsilon(z) \frac{1}{\lambda} \phi\left(\frac{z-z_0}{\lambda}\right) dz, \tag{24}$$

where the integration is a formal notation. For example, expected value $\mathbb{E}\left[\ln \frac{\epsilon_{2^n r}(z_0)}{\epsilon_r(z_0)}\right]$ can be regarded as the comparison between the estimation of local energy dissipation rate $\epsilon(z_0 + z)$ and its rescaled version $\epsilon(z_0 + 2^n z)$ in segment $z \in [-r/2, r/2]$, where we use indicator function $\phi(z) = r^{-1}\mathbb{1}_{[-r/2, r/2]}(z)$ as test function. In terms of the local energy dissipation rate, the scaling law in Eq. (9) means

$$\mathbb{E}\left[\ln \frac{\frac{1}{r}\int_{-r/2}^{r/2} 2^{-\alpha n}\epsilon(z_0 + 2^n z)dz}{\frac{1}{r}\int_{-r/2}^{r/2} \epsilon(z_0 + z)dz}\right] = 0, \tag{25}$$

because

$$\epsilon_r(z_0) = \frac{1}{r}\int\limits_{-r/2}^{r/2} \epsilon(z_0 + z)dz, \tag{}$$

$$\epsilon_{2^n r}(z_0) = \frac{1}{r}\int\limits_{-r/2}^{r/2} \epsilon(z_0 + 2^n z)dz. \tag{26}$$

3. In principle, the scaling property represents the self-similarity underlying the local energy dissipation rate defined along the depth axis.

From Eq. (10), $\mathbb{E}\left[\ln \frac{2^{-\alpha n}\epsilon_{2^n r}(z_k)}{\epsilon_r(z_k)}\right] = 0$, which implies that the sequence of $2^{-\alpha n}\epsilon_{2^n r}(z_k)$ is similar to that of $\epsilon_r(z_k)$ in the logarithmic scale. We can illustrate the self-similarity inherent to the profile data by scaling them appropriately to obtain similar sequences $\zeta_{2^n r}$ in Fig. 12. Specifically, if we use a test function defined with the aid of indicator function

$$\phi_{\langle z_1 \rangle}(z) = r^{-1}\mathbb{1}_{[z_1 - r/2, z_1 + r/2]}(z), \tag{27}$$

and rescale it to

$$\phi_{z_0}^{2^n}(z) = 2^{-n}\phi_{\langle z_1 \rangle}\left(\frac{z-z_0}{2^n}\right), \tag{28}$$

the evaluation of $\epsilon$ against this test function is given by

$$\int \epsilon(z') 2^{-n}\phi_{\langle z_1 \rangle}\left(\frac{z'-z_0}{2^n}\right) dz'$$
$$= \frac{1}{r}\int\limits_{-r/2}^{r/2} \epsilon\left(z_0 + 2^n(z'+z_1)\right) dz'$$
$$= \epsilon_{2^n r}\left(z_0 + 2^n z_1\right), \tag{29}$$

(c) Author(s) 2018. CC BY 4.0 License.



which has scale $2^{\alpha n}$. We can thus define statistically invariant sequence

$$\zeta_{2^n r}(z_0 + z) \equiv 2^{-\alpha n} \epsilon_{2^n r} (z_0 + 2^n z) \tag{30}$$

around $z_0$ for each $n = 0, 1, \cdots$.

## 5   Conclusions

1.  In this paper, we show that the logarithmic sequences of observed depth-averaged energy dissipation rate exhibit a scaling dependence, and formalize the estimation of the population mean for the logarithm of the vertically averaged rate by using this dependence.

2.  From the scaling property in the observed profiles, we found that averaging neighbouring points increases the expected value of its logarithm by a rate proportional to the logarithm of the averaging interval. Furthermore, this property allows us to estimate the population mean for the logarithm of the vertically averaged energy dissipation rate from ocean turbulence profile data.

3.  By scaling up and promoting the observation value at each depth to one corresponding to the whole profile, information from a virtual series of profile observations can be extrapolated from a single profile observation.

4.  The estimate of the vertically averaged energy dissipation rate is accompanied by an explicitly defined estimation error for the population mean, and serves as a measure for distinguishing momentarily large but intermittent values from persistently large energy dissipation on average. In addition, the estimated vertical average should correspond to the energy dissipation rate used in ocean general circulation models, in which case the water column depth can be considered for $\ell$ in Eq. (16).

5.  The scaling exponent depends on the horizontal location, as the turbulence differs among places. Therefore, more observations are needed to establish a more accurate scaling relation. To estimate the spatial distribution of energy dissipation, data assimilation into an ocean general circulation model can be performed using the procedure proposed in this study.

*Data availability.* The ocean turbulence dataset is under preparation for public release by the Atmosphere and Ocean Research Institute, University of Tokyo.





## Appendix A: Measurement error in scaling analysis

If observational data $\epsilon_{\text{obs},r}$ contain measurement error $\eta_r$ obeying $\mathcal{N}(0, \Sigma^2)$, from logarithmic manipulation we obtain

$$
\begin{aligned}
\mathbb{E}\left[\ln \epsilon_{\text{obs},r}\right] &= \lim_{q \to 0+} \mathbb{E}\left[\frac{\epsilon_{\text{obs},r}^q - \epsilon_{\text{obs},r}^{-q}}{2q}\right] \\
&\doteqdot \mathbb{E}\left[\ln \epsilon_r\right] - \frac{1}{2}\mathbb{E}\left[\left(\frac{\eta_r}{\epsilon_r}\right)^2\right].
\end{aligned}
\tag{A1}
$$

This has the following effect on scaling:

$$
\begin{aligned}
&\mathbb{E}\left[\ln \epsilon_{\text{obs},r'} - \ln \epsilon_{\text{obs},r}\right] \\
&\doteqdot \mathbb{E}\left[\ln \epsilon_{r'} - \ln \epsilon_r\right] - \frac{1}{2}\Sigma^2 \mathbb{E}\left[(\epsilon_r)^{-2}\right]\left[\frac{\mathbb{E}\left[(\eta_{r'}/\epsilon_{r'})^2\right]}{\mathbb{E}\left[(\eta_r/\epsilon_r)^2\right]} - 1\right] \\
&= \alpha\left(\ln r' - \ln r\right) + D_r\left[1 - \left(\frac{r'}{r}\right)^{-1+h(-2)}\right],
\end{aligned}
\tag{A2}
$$

where $D_r \equiv \frac{1}{2}\Sigma^2 \mathbb{E}\left[(\epsilon_r)^{-2}\right]$ and $h(-2)$ is the scaling exponent for $\epsilon_r^{-2}$.

The data may contain errors at most in the order of $D_r > 0$. Hence, we use a curve of the form $y = ax + b(1 - e^{-cx})$ for fitting, and regard $c$ as the slope for $\ln\left(\epsilon_{r'}/\epsilon_r\right)$.

*Author contributions.* IY compiled the observational data. SM and SO posed the main problem. IY, SK, SM, and SO helped formulate the hypothesis. NS proposed the method, performed the statistical analyses, and prepared the manuscript with contributions from all co-authors.

*Competing interests.* The authors declare that they have no competing interests.

*Acknowledgements.* The helpful comments from Yutaka Yoshikawa (Kyoto University) are appreciated. This work was partially supported by Grant-in-Aid for Scientific Research on Innovative Areas (MEXT KAKENHI-JP15H05817/JP15H05819). We also thank the members of the project for valuable discussions on the method.





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





**Figure 8.** Scatter plot of depth-averaged energy dissipation rates for upper 2000 m comparing the expected value obtained from rescaled and arithmetic averaging. The top and bottom graphs show the results using a single and three $\widehat{\alpha}$ values, respectively.



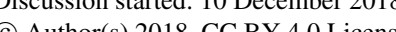

**Figure 9.** Horizontal distribution of vertically averaged energy dissipation rate using a single $\widehat{\alpha}$ value and the rescaled averaging (orange) and arithmetic averaging (blue). The top graph shows the whole evaluated area, and the bottom graph shows an enlarged view around Japan. The units of longitude and latitude are °E and °N, respectively.




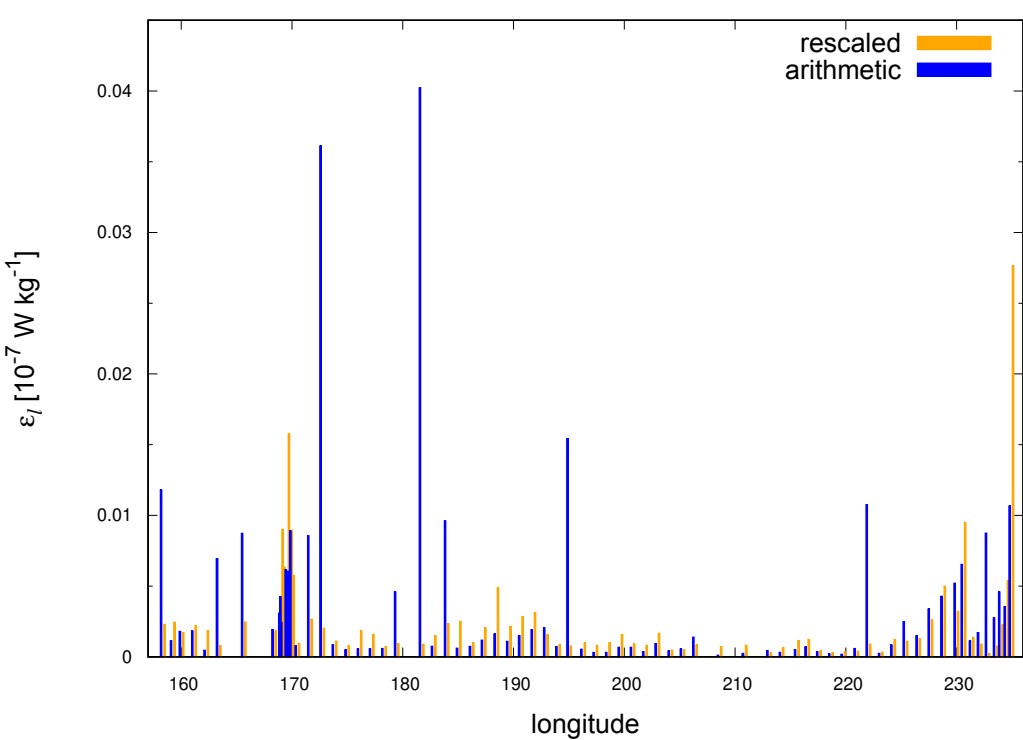

**Figure 10.** Horizontal distribution of vertically averaged energy dissipation rate using a single $\hat{\alpha}$ value and the rescaled averaging (orange) and arithmetic averaging (blue). A cross section along $47\,^{\circ}$N is shown. The unit of longitude is $^{\circ}$E.



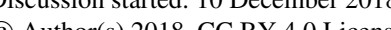



**Figure 11.** Horizontal distribution of vertically averaged energy dissipation rate by using three $\widehat{\alpha}$ values and the rescaled averaging (orange) and arithmetic averaging (blue). The top graph shows the whole evaluated area, and the bottom graph shows an enlarged view around Japan. The units of longitude and latitude are °E and °N, respectively.



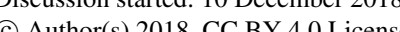

**Figure 12.** Statistical self-similarity in the profiles. Sequences $\left\{ \zeta_{2^n r}(z_0 + z) \equiv 2^{-\alpha n} \epsilon_{2^n r}(z_0 + 2^n z) \mid n = 0, 1, 2, 3 \right\}$ depict arithmetic averages of $2^n$ neighbouring values in the sequence of $\epsilon_r$, shrinking along the $z$ direction by $2^{-n} = 0.5^n$, and then contracting the value by factor $2^{-\alpha n} = 2^{-0.467n} = 0.723^n$. The reference point for scaling transformation is $z_0 = 300$ m.