# Peer review of "Estimating vertically averaged energy dissipation rate"

_Nonlinear Processes in Geophysics, 2018_

## Referee Comment (RC1) · Anonymous Referee #1 · 3 Jan 2019

**Review of *npg-2018-48* "Estimating vertically averaged energy dissipation rates", Sugiura et al.**

It is a commonplace that the global ocean is much less well observed than the atmosphere, a matter of central importance given the role of the oceans in the crucial topic of global heating. It is perhaps for this reason that scaling approaches to the oceans have lagged behind those in use for the atmosphere. Nevertheless, the current work is to be welcomed as being an important new step.

In the atmosphere, things have advanced substantially since Kolmogorov dissipation accompanied by Gaussian and log-normal distributions. In the spirit of trying to encourage the authors to extend and expand their analysis, I suggest a few references herewith:

*S Lovejoy & D Schertzer, 2013, THE WEATHER AND CLIMATE: Emergent Laws and Multifractal Cascades, CUP, ISBN 9781107018983
*A F Tuck, 2010, *Quart. J. R. Meteorol. Soc.,* **136,** 1125-1144. From molecules to meteorology via turbulent scale invariance. Doi: 10.1002/qj.644
* SJ Hovde, AF Tuck, S Lovejoy, D Schertzer - *International Journal of Remote Rensing,* **32,** 5891-5918, 2011. Vertical scaling of temperature, wind and humidity fluctuations: dropsondes from 13 km to the surface of the Pacific Ocean.

It may be that the oceanic dissipation can be calculated simply from the kinetic energy components, but as a non-oceanographer I am bound to ask whether the effects of salinity, acidity and the entropy of mixing have been properly accounted. The atmospheric experience suggests that dissipation must be treated explicitly, even though it is still not an automatic procedure.

Figures 4 and 6 give me pause before recommending this paper for publication. The Gaussian fit in Figure 4 is telling in my opinion: to the left of the maximum, the fit near maximum slope is poor, and the poor fit continues, with the opposite sense, into the tail, which is clearly longer and fatter than its counterpart on the right hand side. Figure 6, in both halves, shows curves that are not sufficiently linear to sustain claims of scaling.

My recommendation is that the authors should be encouraged, with the proviso that they need to meet the criticisms offered above.

---

## Referee Comment (RC2) · Anonymous Referee #2 · 3 Jan 2019

This manuscript discusses the estimation of vertically averaged dissipation in marine turbulence. The authors seem to rediscover some well known results (scale dependence of the local average of the energy dissipation in turbulence), and seem also to ignore the relevant literature, which is vast and classical on this topic.

The manuscript lacks a clear structure; it lacks a review of the literature about intermittent turbulence, and intermittent marine turbulence. The problem addressed is not well explained and globally the whole object of the manuscript does not seem to be to be relevant. I do not suggest to accept such manuscript. I do not recommend major changes: this manuscript must be totally rewritten.

Below some more comments to help improve the manuscript (which should be completely rewritten, taking into account the theoretical framework of intermittent multi fracPrinter-friendly version
tal turbulence): Equation 1: a general book on turbulence should be cited, such as e.g. Pope (2000). Text between equation 1 and equation 3: the authors should indicate that the local energy dissipation in turbulence is intermittent and that an expression such as equation 3 has been proposed by Kolmogorov (1962) to deal with the scale dependence of the locally averaged energy dissipation. Kolmogorov (1941) scaling law should be cited and the scale dependence of the statistics of the locally averaged energy dissipation, given in the framework of multifractal cascade models in turbulence (a relevant reference can be here Frisch 1995) should be provided. It is correct that a lognormal approximation for the dissipation is often assumed, but it is also known that the turbulent dissipation is not strictly lognormal. There are many references on such topic, some of them should be cited.

The authors should discuss the inertial range in which there is a cascade from large to small scales. The scale dependence of the statistics of the locally averaged energy dissipation should be found in the inertial range. In the multifractal framework, which is widely used to describe and model the intermittency of the dissipation, the scale dependence of the moments of the locally averaged dissipation field has a theoretical expression which could be tested in the manuscript.

About the data analyzed: what is the quantity measured? The dissipation epsilon cannot be directly measured. Sometimes epsilon is estimated from vertical profiles using some hypothetical expression: this must be specified and the relevance of the formulae should be discussed. In the inertial range, in the framework of multiplicative cascades models, the dissipation field has a scaling power-law Fourier spectrum. This should be check using the data. The PDF given in Figure 4 is not lognormal, very clearly. It is not symmetric; it has fat tails. A lognormal test can be applied to check the quality of the lognormal fit of the PDF.

The correct average of the dissipation field is the arithmetic average; other types of averages -geometric or taking log- have no physical meaning. This questions the objective of the manuscript, since the authors perform statistics on the log of epsilon, as-
suming Gaussianity of this quantity. Since this assumption is an approximation, what is the quality of the analysis done in this manuscript? The authors should try to quantify this.

NPGD

---

## Author Comment (AC2) · 19 Mar 2019

**Reply to the first reviewer**

The authors sincerely appreciate the first referee's careful review of the manuscript. Taking into account the comments from the two reviewers, we will substantially revise the manuscript to establish consistency with the modern view of intermittency. The authors' responses to the reviewer's comments are as follows:

**General remark** *It is a commonplace that the global ocean is much less well observed than the atmosphere, a matter of central importance given the role of the oceans in the crucial topic of global heating. It is perhaps for this reason that scaling approaches to the oceans have lagged behind those in use for the atmosphere. Nevertheless, the current work is to be welcomed as being an important new step.*

**Reply:**

We appreciate the reviewer's encouragement. We hope this study will serve as one of the important steps in the intermittency study of oceanic turbulence.

**1.** *In the atmosphere, things have advanced substantially since Kolmogorov dissipation accompanied by Gaussian and log-normal distributions. In the spirit of trying to encourage the authors to extend and expand their analysis, I suggest a few references herewith:*

*\*S Lovejoy & D Schertzer, 2013, THE WEATHER AND CLIMATE: Emergent Laws and Multifractal Cascades, CUP, ISBN 9781107018983*

*\*A F Tuck, 2010, Quart. J. R. Meteorol. Soc., 136, 1125-1144. From molecules to meteorology via turbulent scale invariance. Doi: 10.1002/qj.644*

*\* SJ Hovde, AF Tuck, S Lovejoy, D Schertzer - International Journal of Remote Rensing, 32, 5891-5918, 2011. Vertical scaling of temperature, wind and humidity fluctuations: dropsondes from 13 km to the surface of the Pacific Ocean.*

**Reply:**

We appreciate the reviewer for introducing the important references. The literature review was not up to the required standard during the preparation of our paper. Following the reviewer's advice, we have studied several important papers and books on scaling laws and intermittency in meteorology and other applications. As a result, we decided to extend our analysis to a general moment with exponent $q$ and interpret it in the multifractal framework as follows.

In fact, the scale dependence of $\log \epsilon_r$ is just a special case of the scaling of the general $q$-th moment of $\epsilon_r$:

$$\mathbb{E}\left[(\epsilon_r)^q\right] \propto r^{-K(q)}, \tag{1}$$

where $K(q)$ is the scaling exponent introduced in the Introduction. If we take the derivative with respect to $q$ and set $q = 0$, we get

$$\mathbb{E}\left[\log\left(\epsilon_r\right)\right] \propto -K'(0)\log r. \tag{2}$$

For this reason, we perform the scaling analysis (1) of observational data with respect to various moment exponents.

The scalings for several moments are shown in Fig. 1. The observational curve of $(q, K(q))$ in the range of $0 \leq q \leq 2$ is shown in Fig. 2 in cyan. Two types of theoretical expression, log-normal ($K(q) = C_1(q^2 - q)$, red) and log-stable ($K(q) = \frac{C_1}{\alpha - 1}(q^\alpha - q)$, black) cascade models [e.g., Lovejoy and Schertzer, 2013], are tested. While there is some deviation from the log-normal model with $C_1 = 0.316$, the data are well fitted to the log-stable model with the multifractal index $\alpha = 1.59$ and the codimension of the mean $C_1 = 0.357$. These values are largely consistent with previous results for the atmospheric dissipation fields [e.g., Chigirinskaya et al., 1994, Lazarev et al., 1994]. As our data have the sampling dimension $D_s = \frac{\log N_s}{\log (L/r)} \simeq \frac{\log 400}{\log (5000/5)} = 0.867$, the upper bound for $q$ is calculated to be $q_s = 2.95$ (Fig. 3), which justifies the range $0 \leq q \leq 2$.

From these results, we adopt a log-stable generator for the multiplicative cascade of turbulent dissipation.

[Figure]

Figure 1: Scale dependency of the moments: $(\log (r/r_0), -\log \mathbb{E}\left[(\epsilon_r/\epsilon_{r_0})^q\right])$.    $K(0.5) = -0.103 \pm 0.006$, $K(1.5) = 0.241 \pm 0.036$, $K(2.0) = 0.616 \pm 0.053$.

[Figure]

Figure 2: Moment scaling exponent $K(q)$ for observational data (cyan), the best-fitting log-normal model ($C_1 = 0.316$, $\alpha = 2$; red), and the best-fitting log-stable model ($C_1 = 0.357$, $\alpha = 1.59$; black).

[Figure]

Figure 3: Codimension $c(\gamma)$ of singularities $\gamma$ for the log-stable model (black) and for the log-normal model (red). The sampling dimension $D_s$ and the limitation for the moment exponent (the slope of the navy line) are also shown.

**2.** *It may be that the oceanic dissipation can be calculated simply from the kinetic energy components, but as a non-oceanographer I am bound to ask whether the effects of salinity, acidity and the entropy of mixing have been properly accounted. The atmospheric experience suggests that dissipation must be treated explicitly, even though it is still not an automatic procedure.*

**Reply:**

As the reviewer pointed out, the paper lacked an explanation regarding how the energy dissipation rate has been calculated from micro-scale temperature. We are going to include the following description in the Method section.

Turbulent energy dissipation rates $\epsilon$ were estimated as follows. Micro-scale temperature fields were observed using the fast-response Fasttip Probe model 07 (FP07) thermistors attached to frames measuring conductivity, temperature, and depth (CTD) as common oceanographic observational platforms. $\epsilon$ was derived by detecting the Batchelor wavenumber [Batchelor, 1959] with fitting [Ruddick et al., 2000] theoretical spectrum [Kraichnan, 1968] to the observed temperature vertical gradient spectra after correcting the spectra with a double-pole function and a 3-ms time-constant [Goto et al., 2016]. Each data was evaluated for a depth interval of approximately 10 m with a half overlap, to yield 5-dbar interval data. We herein include all data without any quality screening so as not to miss the extreme values, which are important for the purpose of investigating intermittency.

**3.** *Figures 4 and 6 give me pause before recommending this paper for publication. The Gaussian fit in Figure 4 is telling in my opinion: to the left of the maximum, the fit near maximum slope is poor, and the poor fit continues, with the opposite sense, into the tail, which is clearly longer and fatter than its counterpart on the right hand side. Figure 6, in both halves, shows curves that are not sufficiently linear to sustain claims of scaling.*

**Reply:**

We appreciate the reviewer for letting us know of these crucial points. As pointed out by the reviewer, indeed Fig. 4 is different from log-normal. We are going to treat it as a log-stable distribution in the revised text as follows.

Regarding Fig. 4, a closer look at it reveals that the logarithmic values do not follow Gaussian statistics. Although the poor fit for the small values (left-hand-side tail) may be partly caused by the measuring errors, the fat tail should probably come from the non-Gaussian nature of the data. The poor fit for the large values (right-hand-side tail) is even more problematic because it directly affects the appearance frequency of extreme values. The asymmetry and the mismatched tail behavior from Gaussian in the histogram thus suggest that the generator for multiplicative processes in this dissipation field does not obey Gaussian statistics.

In the framework of multifractal cascade [e.g., Schertzer et al., 1995, Mandelbrot and Evertsz, 1996, Lovejoy and Schertzer, 2013], the generator should obey a left-sided stable distribution. As energy dissipation appears as the exponential of the generator, the critical point in the comparison between the statistical model and the observational data is the tail structure for the large values (right-hand-side tail). Figure 4 is the histogram for observed data in the log–log scale with a Gaussian distribution and a stable distribution for reference. The rare occurrence of singular events in the right-hand-side tail is better expressed by the stable distribution (we will discuss the sample generation procedure later). For this reason, we adopt a multiplicative cascade with a generator that obeys a stable distribution.

[Figure]

Figure 4: Distribution of observational data in log-log scale (cyan), the best-fitting Gaussian distribution (red), and the statistics of samples generated from a log-stable multiplicative cascade (black).

Regarding Fig. 6, the insufficient linearity in the scaling curves should come from the logarithm of $\epsilon$ as it tends to suffer from observational noise, as mentioned in the Appendix of the previous manuscript. Rather than stick to a specific moment for scaling, a better way of performing the scaling analysis would be extending it to the moment with an arbitrary degree, as written in the reply to comment #1. We have investigated the scale dependency of the moments with exponent $0 \leq q \leq 2$ (Fig 1). We think the linearity is acceptable for claiming scaling this time.

**References**

G. K. Batchelor. Small-scale variation of convected quantities like temperature in turbulent fluid Part 1. General discussion and the case of small conductivity. *J. Fluid Mech.*, 5(1):113–133, 1959. doi: 10.1017/S002211205900009X.

Y. Chigirinskaya, D. Schertzer, S. Lovejoy, A. Lazarev, and A. Ordanovich. Unified multifractal atmospheric dynamics tested in the tropics: part i, horizontal scaling and self criticality. *Nonlinear Processes in Geophysics*, 1(2/3):105–114, 1994. doi: 10.5194/npg-1-105-1994. URL https://www.nonlin-processes-geophys.net/1/105/1994/.

Y. Goto, I. Yasuda, and M. Nagasawa. Turbulence Estimation Using Fast-Response Thermistors Attached to a Free-Fall Vertical Microstructure Profiler. *Journal of Atmospheric and Oceanic Technology*, 33(10):2065–2078, 2016. doi: 10.1175/JTECH-D-15-0220.1.

R. H. Kraichnan. Small-scale structure of a scalar field convected by turbulence. *The Physics of Fluids*, 11(5):945–953, 1968.

A. Lazarev, D. Schertzer, S. Lovejoy, and Y. Chigirinskaya. Unified multifractal atmospheric dynamics tested in the tropics: part ii, vertical scaling and generalized scale invariance. *Nonlinear Processes in Geophysics*, 1(2/3):115–123, 1994. doi: 10.5194/npg-1-115-1994. URL https://www.nonlin-processes-geophys.net/1/115/1994/.

S. Lovejoy and D. Schertzer. *The Weather and Climate: Emergent Laws and Multifractal Cascades*. Cambridge University Press, 2013.

B. B. Mandelbrot and C. J. G. Evertsz. *Exactly Self-similar Left-sided Multifractals*, pages 367–399. Springer Berlin Heidelberg, Berlin, Heidelberg, 1996. ISBN 978-3-642-84868-1.

B. Ruddick, A. Anis, and K. Thompson. Maximum Likelihood Spectral Fitting: The Batchelor Spectrum. *Journal of Atmospheric and Oceanic Technology*, 17(11):1541–1555, 2000.

D. Schertzer, S. Lovejoy, and F. Schmitt. Structures in turbulence and multifractal universality. In *Small-Scale Structures in Three-Dimensional Hydrodynamic and Magnetohydrodynamic Turbulence*, pages 137–144. Springer, 1995.

**Reply to the second reviewer**

The authors sincerely appreciate the second referee's careful review of the manuscript. Taking into account the comments from the two reviewers, we will substantially revise the manuscript to establish the consistency with the modern view of intermittency. The authors' responses to the reviewer's comments are as follows:

**1.** *The authors seem to rediscover some well known results (scale dependence of the local average of the energy dissipation in turbulence), and seem also to ignore the relevant literature, which is vast and classical on this topic.*

**Reply:**

As the reviewer pointed out, we did not review enough relevant literature. We are going to include a review of the scaling study of turbulence and stress on the universal multifractal theory in the Introduction as follows.

In fully developed turbulence, there exists the inertial subrange where the advective term is dominant to the molecular viscosity term in the Navier–Stokes equation [e.g., Pope, 2000]. In the inertial subrange, there is a cascade of energy from large to small [e.g., Richardson, 1922]. As the first quantitative theory on the energy cascade, Kolmogorov [1941] established a relationship that said velocity fluctuations are locally isotropic and determined by the homogeneous energy dissipation rate:

$$\mathbb{E}\left[|v(x+\ell) - v(x)|\right] \approx \epsilon^{1/3}\ell^{1/3}. \tag{1}$$

Soon after, it is criticized that energy dissipation rate is not homogeneous but shows significant random fluctuations. To address this issue, he published a refined theory [Kolmogorov, 1962] that stated that i) $\log \epsilon_r$, the logarithm of the spatially averaged energy dissipation rate over a scale $r$, obeys a Gaussian distribution, and ii) its mean and variance are dependent on scale $\log \frac{L}{r}$ ($L$: the outer scale) with the proportionality constants $-\mu_1/2$ and $\mu_1$, respectively. This implies that the multiplicative cascade comprises underlying iid generators $\Gamma_j$, and $\log \epsilon_r$ is expressed as a random walk with step sizes $\Gamma_j$ [e.g., Yaglom, 1966]:

$$\log \frac{\epsilon_r}{\epsilon_L} = \sum_{j=1}^{n} \log \frac{\epsilon_{re^{(j-1)h}}}{\epsilon_{re^{jh}}} \equiv \sum_{j=1}^{n} \Gamma_j. \tag{2}$$

where $h = \frac{1}{n}\log\frac{L}{r}$ is the resolution increment. Then, an application of the central limit theorem:

$$\sum_{j=1}^{n} \frac{\Gamma_j + \frac{\mu_1}{2}h}{(\mu_1 nh)^{1/2}} \sim \mathcal{N}(0,1), \quad \text{as } n \to \infty \tag{3}$$

implies that if $\Gamma_j$ are iid random variables, each of which having mean $-\mu_1 h/2$ and variance $\mu_1 h$, then $\log\frac{\epsilon_r}{\epsilon_L}$ obeys $\mathcal{N}(-\frac{\mu_1}{2}nh, \mu_1 nh)$, which is consistent with the statement of Kolmogorov [1962]. Although this log-normal cascade well describes turbulence in an approximate sense, there are some shortcomings at high-order moments and in the application of various fluid phenomena.

There have been several alternative theories for multiplicative cascades [e.g, Frisch, 1995]. Among them, the universal multiplicative cascade model [e.g., Lovejoy and Schertzer, 2013] is the most

promising theory that explains vast phenomena, including turbulence, other geophysical phenomena, and several fractal-like appearances in natural and even man-made objects.

The basic mechanism of multifractality is encoded in the codimension setting: $\epsilon_r$ that has the dominant contribution to a moment occupies a corresponding subdomain, whose size can vary according to the moment exponent [Stanley and Meakin, 1988, Lovejoy and Schertzer, 2013]. Suppose the $q$-th moment at scale $r$ is described as

$$Z_r(q) = \sum_{\epsilon_r} e^{F(\epsilon_r)}; \quad F(\epsilon_r) = \log n(\epsilon_r) + q \log \epsilon_r, \tag{4}$$

where $n(\epsilon_r)$ is the probability distribution of $\epsilon_r$, and we avoid using integral formulation for readability. We assume that the value is concentrated on the most dominant term at $\epsilon_r^*$:

$$Z_r(q) \approx e^{F(\epsilon_r^*)}. \tag{5}$$

Define the singularity $\gamma$ and its codimension $c(\gamma)$ at that point as

$$\epsilon_r^* \approx \left(\frac{L}{r}\right)^{\gamma}; \quad n(\epsilon_r^*) \approx \left(\frac{L}{r}\right)^{-c(\gamma)}. \tag{6}$$

The exponents $\gamma$ and $c(\gamma)$ indicate how extreme and how rare the value of $\epsilon_r$ that dominate the $q$-the moment is, respectively. The right-hand-side tail of the probability distribution is thus determined by the exponent $c(\gamma)$, which can be different from a quadratic function if the generator obeys a stable distribution with $\alpha < 2$ [Eq. (1.2.11) of Samorodnitsky and Taqqu, 1994].

Substituting Eq. (5) into Eq. (4), we get

$$K(q) \equiv \frac{\log Z_r(q)}{\log(L/r)} = \max_{\gamma}(\gamma q - c(\gamma)). \tag{7}$$

From the extremum condition, we have

$$\frac{dc}{d\gamma} = q; \quad \frac{dK}{dq} = \gamma. \tag{8}$$

Hence, $K(q)$ and $c(\gamma)$ are the Legendre transforms of each other:

$$c(\gamma) = \max_{q}(\gamma q - K(q)). \tag{9}$$

A crucial point in universal multifractal is that the generator of the cascade obeys a stable distribution, which is a generalization of Gaussian distribution, and has the following characteristic function:

$$\Gamma_j \sim S_\alpha(\sigma, \beta, \mu) \iff \mathbb{E}[e^{i\theta\Gamma_j}] = \exp\left\{-\sigma^\alpha|\theta|^\alpha\left(1 - i\beta(\text{sign}\theta)\tan\frac{\pi\alpha}{2}\right) + i\mu\theta\right\}, \tag{10}$$

with $0 < \alpha \le 2$ and $\alpha \ne 1$ [Definition 1.1.6 of Samorodnitsky and Taqqu, 1994].

Accordingly, the construction of random walk in the cascade is built on the basis of a generalized version of central limit theorem: if $\Gamma_j$ are iid random variables, each of which obeys $S_\alpha(\sigma h^{1/\alpha}, \beta, \mu h)$, then as $n \to \infty$ we have

$$\sum_{j=1}^{n} \frac{\Gamma_j - \mu h}{\sigma(nh)^{1/\alpha}} \sim S_\alpha(1, \beta, 0); \quad \sum_{j=1}^{n} \Gamma_j \sim S_\alpha(\sigma(nh)^{1/\alpha}, \beta, \mu(nh)), \tag{11}$$

where $h = \frac{1}{n} \log \frac{L}{r}$ is the resolution increment.

If we set $\beta = -1$ and $\mu = -\widehat{\sigma_\alpha}^\alpha \equiv -\sigma^\alpha \left( \cos \frac{\pi}{2} (2 - \alpha) \right)^{-1}$, then $\mathrm{e}^{\Gamma_j}$ satisfies energy conservation and has finite moments [Proposition 1.2.12 of Samorodnitsky and Taqqu, 1994]:

$$\mathbb{E}\left[ \mathrm{e}^{q\Gamma_j} \right] = \mathrm{e}^{K(q)h}; \quad \mathbb{E}\left[ \mathrm{e}^{q \sum_{j=1}^n \Gamma_j} \right] = \mathrm{e}^{K(q)nh}, \tag{12}$$

where $K(q) \equiv \widehat{\sigma_\alpha}^\alpha (q^\alpha - q)$.

From the above, spatially averaged energy dissipation rate can be expressed as

$$\epsilon_r = \bar{\epsilon} \exp \left( \sum_{j=1}^n \Gamma_j \right), \tag{13}$$

where $\Gamma_j$ is the same as in Eq. (11), and $\bar{\epsilon}$ is the energy input at the outer scale $L$, which is conserved through the cascade in a probabilistic sense. At a fixed scale $r$, $\epsilon_r(x)$ in the space domain are not independent but correlated [e.g., Schmitt, 2003]:

$$\mathbb{E}[\epsilon_r(x)\epsilon_r(x + \ell)] \propto \ell^{-K(2)}, \quad \text{for } \ell > r. \tag{14}$$

**2.** *The manuscript lacks a clear structure; it lacks a review of the literature about intermittent turbulence, and intermittent marine turbulence. The problem addressed is not well explained and globally the whole object of the manuscript does not seem to be to be relevant. I do not suggest to accept such manuscript. I do not recommend major changes: this manuscript must be totally rewritten.*

**Reply:**

Taking seriously into account the reviewer's critical comment, we will entirely rewrite the manuscript to present a meaningful result regarding the analysis of intermittency and the estimation of mean value based on the multifractal framework. For reviewing the intermittency of turbulence, please refer to the reply to comment #1. For the averaging method based on multifractal multiplicative cascade, please refer to our reply to comment #6.

**3.** *Equation 1: a general book on turbulence should be cited, such as e.g. Pope (2000). Text between equation 1 and equation 3: the authors should indicate that the local energy dissipation in turbulence is intermittent and that an expression such as equation 3 has been proposed by Kolmogorov (1962) to deal with the scale dependence of the locally averaged energy dissipation. Kolmogorov (1941) scaling law should be cited and the scale dependence of the statistics of the locally averaged energy dissipation, given in the framework of multifractal cascade models in turbulence (a relevant reference can be here Frisch 1995) should be provided. It is correct that a lognormal approximation for the dissipation is often assumed, but it is also known that the turbulent dissipation is not strictly lognormal. There are many references on such topic, some of them should be cited.*

**Reply:**

A review on turbulence intermittency will be added to the manuscript. The relevant literature is cited therein. Please see the reply to comment #1.

**4.** *The authors should discuss the inertial range in which there is a cascade from large to small scales. The scale dependence of the statistics of the locally averaged energy dissipation should be found in the inertial range. In the multifractal framework, which is widely used to describe and model the intermittency of the dissipation, the scale dependence of the moments of the locally averaged dissipation field has a theoretical expression which could be tested in the manuscript.*

**Reply:**

The explanation for the cascade models in the inertial range will be added to the introduction. Please refer to the reply to comment #1. Regarding the scale dependency of the moments, the following explanation will be added.

In fact, the scale dependence of $\log \epsilon_r$ is just a special case of the scaling of the general $q$-th moment of $\epsilon_r$:

$$\mathbb{E}\left[(\epsilon_r)^q\right] \propto r^{-K(q)}, \tag{15}$$

where $K(q)$ is the scaling exponent introduced in the Introduction. If we take the derivative with respect to $q$ and set $q = 0$, we get

$$\mathbb{E}\left[\log\left(\epsilon_r\right)\right] \propto -K'(0)\log r. \tag{16}$$

For this reason, we perform the scaling analysis (15) of observational data with respect to various moment exponents.

The scalings for several moments are shown in Fig. 1. The observational curve of $(q, K(q))$ in the range of $0 \leq q \leq 2$ is shown in Fig. 2 in cyan. Two types of theoretical expression, log-normal ($K(q) = C_1(q^2 - q)$, red) and log-stable ($K(q) = \frac{C_1}{\alpha-1}(q^\alpha - q)$, black) cascade models [e.g., Lovejoy and Schertzer, 2013], are tested. While there is some deviation from the log-normal model with $C_1 = 0.316$, the data are well fitted to the log-stable model with the multifractal index $\alpha = 1.59$ and the codimension of the mean $C_1 = 0.357$. These values are largely consistent with previous results for the atmospheric dissipation fields [e.g., Chigirinskaya et al., 1994, Lazarev et al., 1994]. As our data have the sampling dimension $D_s = \frac{\log N_s}{\log(L/r)} \simeq \frac{\log 400}{\log(5000/5)} = 0.867$, the upper bound for $q$ is calculated to be $q_s = 2.95$ (Fig. 3), which justifies the range $0 \leq q \leq 2$.

From these results, we adopt a log-stable generator for the multiplicative cascade of turbulent dissipation.

[Figure]

Figure 1: Scale dependency of the moments: $(\log{(r/r_0)}, -\log{\mathbb{E}\left[(\epsilon_r/\epsilon_{r_0})^q\right]})$. $K(0.5) = -0.103 \pm 0.006$, $K(1.5) = 0.241 \pm 0.036$, $K(2.0) = 0.616 \pm 0.053$.

[Figure]

Figure 2: Moment scaling exponent $K(q)$ for observational data (cyan), the best-fitting log-normal model ($C_1 = 0.316$, $\alpha = 2$; red), and the best-fitting log-stable model ($C_1 = 0.357$, $\alpha = 1.59$; black).

[Figure]

Figure 3: Codimension $c(\gamma)$ of singularities $\gamma$ for the log-stable model (black) and for the log-normal model (red). The sampling dimension $D_s$ and the limitation for the moment exponent (the slope of the navy line) are also shown.

**5.** *About the data analyzed: what is the quantity measured? The dissipation epsilon cannot be directly measured. Sometimes epsilon is estimated from vertical profiles using some hypothetical expression: this must be specified and the relevance of the formulae should be discussed. In the inertial range, in the framework of multiplicative cascades models, the dissipation field has a scaling power-law Fourier spectrum. This should be check using the data. The PDF given in Figure 4 is not lognormal, very clearly. It is not symmetric; it has fat tails. A lognormal test can be applied to check the quality of the lognormal fit of the PDF.*

**Reply:**

As the reviewer pointed out, the paper lacked explanation regarding how the energy dissipation rate has been calculated from micro-scale temperature. We will add the following description in the Methods section. Please refer to the corresponding papers for the treatment of the spectrum.

Turbulent energy dissipation rates $\epsilon$ were estimated as follows. Micro-scale temperature fields were observed using the fast-response Fasttip Probe model 07 (FP07) thermistors attached to frames measuring conductivity, temperature, and depth (CTD) as common oceanographic observational platforms. $\epsilon$ was derived by detecting the Batchelor wavenumber [Batchelor, 1959] with fitting [Ruddick et al., 2000] theoretical spectrum [Kraichnan, 1968] to the observed temperature vertical gradient spectra after correcting the spectra with a double-pole function and a 3-ms time-constant [Goto et al., 2016]. Each data was evaluated for a depth interval of approximately 10 m with a half overlap, to yield 5-dbar interval data. We herein include all data without any quality screening so as not to miss the extreme values, which are important for the purpose of investigating intermittency.

As pointed out by the reviewer, the PDF does not obey log-normal statistics. We are going to treat it as a log-stable distribution in the revised manuscript as follows.

Regarding Fig. 4, a closer look at it reveals that the logarithmic values do not follow Gaussian statistics. Although the poor fit for the small values (left-hand-side tail) may be partly caused by the measuring errors, the fat tail should probably come from the non-Gaussian nature of the data. The poor fit for the large values (right-hand-side tail) is even more problematic because it directly affects the appearance frequency of extreme values. The asymmetry and the mismatched tail behavior from Gaussian in the histogram thus suggest that the generator for multiplicative processes in this dissipation field does not obey Gaussian statistics.

In the framework of multifractal cascade [e.g., Schertzer et al., 1995, Mandelbrot and Evertsz, 1996, Lovejoy and Schertzer, 2013], the generator should obey a left-sided stable distribution. As energy dissipation appears as the exponential of the generator, the critical point in the comparison between the statistical model and the observational data is the tail structure for the large values (right-hand-side tail). Figure 4 is the histogram for observed data in the log–log scale with a Gaussian distribution and a stable distribution for reference. The rare occurrence of singular events in the right-hand-side tail is better expressed by the stable distribution (we will discuss the sample generation procedure later). For this reason, we adopt a multiplicative cascade with a generator that obeys a stable distribution.

[Figure]

Figure 4: Distribution of observational data in log-log scale (cyan), the best-fitting Gaussian distribution (red), and the statistics of samples generated from a log-stable multiplicative cascade (black).

**6.** *The correct average of the dissipation field is the arithmetic average; other types of averages -geometric or taking log- have no physical meaning. This questions the objective of the manuscript, since the authors perform statistics on the log of epsilon, assuming Gaussianity of this quantity. Since*

*this assumption is an approximation, what is the quality of the analysis done in this manuscript? The authors should try to quantify this.*

**Reply:**

As the reviewer pointed out, the arithmetic average is the primary object for characterizing the mean state of the dissipation field. Meanwhile, oceanographers often use the logarithm of $\epsilon$ to visualize the dissipation field and/or compare the field with parameterization schemes [e.g., Scheifele et al., 2018, Whalen et al., 2015, Waterhouse et al., 2014, Cuypers et al., 2012, Gargett, 1999, Smyth et al., 1997]. One reason is that the logarithmic field has a spatially correlated structure, whose shape is relatively easy to recognize, whereas the original field has a highly intermittent structure. For such practical purposes, we think that the logarithmic field is still of importance. We are going to revise the manuscript regarding the averaging method as follows. This estimation is a simple application of the cascade simulation and will help us avoid the misreading of the logarithmic fields.

To examine the statistical property of geometric averages, we have constructed a simulation model for the multiplicative cascade following the procedure in Schmitt [2003]. It is based on a generator that obeys a left-sided stable distribution $S_\alpha(\sigma h^{1/\alpha}, -1, -\widehat{\sigma_\alpha}^\alpha h)$ with $h \equiv \log 2$, $\widehat{\sigma_\alpha}^\alpha \equiv \sigma^\alpha / \cos \frac{\pi}{2}(2-\alpha) = C_1/(\alpha-1)$ [e.g., Samorodnitsky and Taqqu, 1994].

Consider a fixed horizontal position. The cascade simulation is performed for the variable $X_{i,j}$ with spatial index $1 \leq i \leq 2^N$, and scale index $0 \leq j \leq N$, where $N = \log_2 \frac{L}{r}$.

1. For each spatial index $i = 1, 2, \cdots, 2^N$, set $X_{i,0} = 0$.

2. For each scale index $j = 1, \cdots, N$, repeat the following.

    - For each spatial block $k = 1, 2, \cdots, 2^j$, do the following.

        (a) Generate a random variable $\xi_{k,j}$ that obeys $S_\alpha(1, -1, 0)$ [Misiorek and Weron, 2012].
        (b) For each spatial index $i = (k-1) \cdot 2^{N-j} + 1, \cdots, k \cdot 2^{N-j}$, downscale $X$ by

$$X_{i,j} = X_{i,j-1} - \widehat{\sigma_\alpha}^\alpha h + \sigma h^{\frac{1}{\alpha}} \xi_{k,j}.$$

Using the simulated variable $X_{i,j}$, the energy dissipation rate at the horizontal position $\vec{x}$ and the vertical position $z_i = i r_j$ at resolution $r_j = L/2^j$ is written as

$$\epsilon_{r_j}(\vec{x}, z_i) = \bar{\epsilon}(\vec{x}) \exp(X_{i,j}), \tag{17}$$

where $\bar{\epsilon}(\vec{x})$ denotes the energy input rate at the horizontal position.

What we want to estimate is the energy input rate $\bar{\epsilon}(\vec{x})$ at the horizontal position $\vec{x}$ of the profile observation. Using the above model, here we calculate the following quantities by sampling the realizations of the vertical profile.

The expected value for the geometric average for the neighboring $m$ points starting from an outer condition $\bar{\epsilon}$ for the cascade is

$$a_g(m) = \mathbb{E}\left[\exp\left(\frac{1}{m}\sum_{i=1}^{m}\gamma_i\right)\bigg|\bar{\epsilon}\right]. \tag{18}$$

where $\gamma_i \equiv \log \epsilon_i$, and $\epsilon_i$'s denote the energy dissipation rates for the observed length scale.

As the ratio of the expected values for geometric average to arithmetic average $\Phi(m) \equiv \frac{a_g(m)}{\bar{\epsilon}}$ is independent of the initial condition $\bar{\epsilon}$, we can define an estimator for the average energy $\bar{\epsilon}$ as

$$e_g(m) = \frac{1}{\Phi(m)} \exp\left(\frac{1}{m} \sum_{i=1}^{m} \gamma_i\right). \tag{19}$$

The expected value of $e_g(m)$ is $\bar{\epsilon}$, which means it is an unbiased estimator for the outer condition. The relative error $\eta$ for the estimator $e_g(m)$ is

$$\eta(e_g(m)) = \frac{\sqrt{\mathrm{var}\left[\frac{1}{\Phi(m)} \exp\left(\frac{1}{m} \sum_{i=1}^{m} \gamma_i\right) \Big| \bar{\epsilon}\right]}}{\mathbb{E}\left[\frac{1}{\Phi(m)} \exp\left(\frac{1}{m} \sum_{i=1}^{m} \gamma_i\right) \Big| \bar{\epsilon}\right]} = \frac{\sqrt{\mathrm{var}\left[\exp\left(\frac{1}{m} \sum_{i=1}^{m} \gamma_i\right) \Big| \bar{\epsilon}\right]}}{\mathbb{E}\left[\exp\left(\frac{1}{m} \sum_{i=1}^{m} \gamma_i\right) \Big| \bar{\epsilon}\right]}, \tag{20}$$

while that for arithmetic average $e_a(m) \equiv \frac{1}{m} \sum_{i=1}^{m} \epsilon_i$ is

$$\eta(e_a(m)) = \frac{\sqrt{\mathrm{var}\left[\frac{1}{m} \sum_{i=1}^{m} \epsilon_i \Big| \bar{\epsilon}\right]}}{\bar{\epsilon}}. \tag{21}$$

We calculate these quantities by using the the simulation model for the multiplicative cascade. The result is shown in Fig. 5. It is obvious that the relative error for $e_g(m)$ is comparable to that for $e_a(m)$. For this reason, we propose that geometric averages can also be used for estimating the average energy $\bar{\epsilon}$, provided that it is properly scaled up by the factor $1/\Phi(m)$.

The usage of Fig. 5 is as follows. For example, if you take the geometric average of 128–(512) neighboring points, the population mean $\bar{\epsilon}$ is expected to be $1/0.116 = 8.62$ $(1/0.0515 = 19.4)$ times larger than the geometric average, black curve at $x = 128$ (512), but after rescaling the geometric average should be almost as accurate estimation of $\bar{\epsilon}$ as the arithmetic average, provided that the multiplicative cascade model is accepted.

[Figure]

Figure 5: The expected values of the arithmetic mean (red), which remains constant $\bar{\epsilon} = 1$, and of the geometric mean (black) according to the number of neighboring points. The relative errors for both are also shown (blue and magenta, respectively).

We think there is another way to estimate $\bar{\epsilon}$ in the logarithmic space, which is data assimilation using the cascade model and the observation. As a future direction for the research, we are going to mention a procedure of data assimilation as follows.

We can also estimate the energy input rate $\bar{\epsilon}$ by data assimilation into the energy cascade model. We denote here the logarithm of the energy input rate as $\bar{\gamma} = \log \bar{\epsilon}$, the logarithm of the observed energy dissipation rates as $\gamma_j = \log \epsilon_j$, $j = 1, 2, \cdots, 2^n$, and the logarithm of arithmetic average of the observed energy dissipation rates as $\hat{\gamma} = \log \left( \frac{1}{2^n} \sum_{j=1}^{2^n} \epsilon_j \right)$. We also denote the generators (iid stable random variables) of the $n$-step cascade model as

$$\Gamma = \{\Gamma_{ij} | \, i = 1, 2, \cdots, n; j = 1, 2, \cdots, 2^i\},$$

and its subset as

$$\Gamma' = \{\Gamma_{ij} | \, i = 1, 2, \cdots, n-1; j = 1, 2, \cdots, 2^i\}.$$

Using these generators we can calculate the model counterpart of $\gamma_j$ as

$$\gamma_j^{\mathrm{M}} = \bar{\gamma} + \sum_{i=1}^{n} \Gamma_{i, [(j-1)/2^{n-i}]+1},$$

where $[\cdot]$ is Gauss's symbol. The expected value of $\bar{\gamma}$ under the condition of $\gamma_j$'s is written using Bayes's rule as

$$\mathbb{E}\left[\bar{\gamma}|\{\gamma_j\}\right] = \sum_{\bar{\gamma}} \bar{\gamma} \mathbb{P}\left(\bar{\gamma}|\{\gamma_j\}\right) = \sum_{\bar{\gamma}} \bar{\gamma} \frac{\mathbb{P}\left(\bar{\gamma}, \{\gamma_j\}\right)}{\sum_{\bar{\gamma}} \mathbb{P}\left(\bar{\gamma}, \{\gamma_j\}\right)} = \frac{\sum_{\bar{\gamma}} \bar{\gamma} \mathbb{P}(\bar{\gamma}) \mathbb{P}\left(\{\gamma_j\}|\bar{\gamma}\right)}{\sum_{\bar{\gamma}} \mathbb{P}(\bar{\gamma}) \mathbb{P}\left(\{\gamma_j\}|\bar{\gamma}\right)}, \tag{22}$$

where $\mathbb{P}\left(\{\gamma_j\}|\overline{\gamma}\right) = \mathbb{P}\left(\Gamma'\right)\prod_{j=1}^{2^n}\mathbb{P}\left(\gamma_j - \overline{\gamma} - \sum_{i=1}^{n-1}\Gamma_{i,[(j-1)/2^{n-i}]+1}\right)$.

Regarding $\mathbb{P}\left(\overline{\gamma}\right)$ and $\mathbb{P}\left(\Gamma'\right)$, we can draw the samples of $\overline{\gamma}$ such that

$$\widehat{\gamma} - \overline{\gamma} \sim S_\alpha(\sigma h^{1/\alpha}, -1, -\widehat{\sigma_\alpha}^\alpha h),$$

and that of $\Gamma'$ such that

$$\Gamma_{ij} \sim S_\alpha(\sigma h^{1/\alpha}, -1, -\widehat{\sigma_\alpha}^\alpha h), \quad i = 1, 2, \cdots, n-1; j = 1, 2, \cdots, 2^i.$$

The probability density $\mathbb{P}\left(\gamma_j - \overline{\gamma} - \sum_{i=1}^{n-1}\Gamma_{i,[(j-1)/2^{n-i}]+1}\right)$ can be calculated by using an approximation formula for the $\alpha$-stable distribution.

We thus obtain the ensemble calculation formula for the expected value:

$$\mathbb{E}\left[\overline{\gamma}|\{\gamma_j\}\right] \doteqdot \frac{\sum_{\overline{\gamma},\Gamma'}^{\text{ens}} \overline{\gamma} \prod_{j=1}^{2^n} \mathbb{P}\left(\gamma_j - \overline{\gamma} - \sum_{i=1}^{n-1}\Gamma_{i,[(j-1)/2^{n-i}]+1}\right)}{\sum_{\overline{\gamma},\Gamma'}^{\text{ens}} \prod_{j=1}^{2^n} \mathbb{P}\left(\gamma_j - \overline{\gamma} - \sum_{i=1}^{n-1}\Gamma_{i,[(j-1)/2^{n-i}]+1}\right)}, \tag{23}$$

where $\sum^{\text{ens}}$ denotes the sum over ensemble members. The variance is also calculated by using the similar formula.

Note that a more sophisticated data assimilation method than a naive sampling method will be needed when you treat the generator space with a large dimension size.

**References**

G. K. Batchelor. Small-scale variation of convected quantities like temperature in turbulent fluid Part 1. General discussion and the case of small conductivity. *J. Fluid Mech.*, 5(1):113–133, 1959. doi: 10.1017/S002211205900009X.

Y. Chigirinskaya, D. Schertzer, S. Lovejoy, A. Lazarev, and A. Ordanovich. Unified multifractal atmospheric dynamics tested in the tropics: part i, horizontal scaling and self criticality. *Nonlinear Processes in Geophysics*, 1(2/3):105–114, 1994. doi: 10.5194/npg-1-105-1994. URL https://www.nonlin-processes-geophys.net/1/105/1994/.

Y. Cuypers, P. Bouruet-Aubertot, C. Marec, and J.-L. Fuda. Characterization of turbulence from a fine-scale parameterization and microstructure measurements in the mediterranean sea during the boum experiment. *Biogeosciences*, 9(8):3131–3149, 2012. doi: 10.5194/bg-9-3131-2012. URL https://www.biogeosciences.net/9/3131/2012/.

U. Frisch. *Turbulence: the legacy of AN Kolmogorov*. Cambridge university press, 1995.

A. E. Gargett. Velcro measurement of turbulence kinetic energy dissipation rate $\epsilon$. *Journal of Atmospheric and Oceanic Technology*, 16(12):1973–1993, 1999.

Y. Goto, I. Yasuda, and M. Nagasawa. Turbulence Estimation Using Fast-Response Thermistors Attached to a Free-Fall Vertical Microstructure Profiler. *Journal of Atmospheric and Oceanic Technology*, 33(10):2065–2078, 2016. doi: 10.1175/JTECH-D-15-0220.1.

A. N. Kolmogorov. The local structure of turbulence in incompressible viscous fluid for very large Reynolds numbers. *Cr Acad. Sci. URSS*, 30:301–305, 1941.

A. N. Kolmogorov. A refinement of previous hypotheses concerning the local structure of turbulence in a viscous incompressible fluid at high Reynolds number. *Journal of Fluid Mechanics*, 13(1): 82–85, 1962.

R. H. Kraichnan. Small-scale structure of a scalar field convected by turbulence. *The Physics of Fluids*, 11(5):945–953, 1968.

A. Lazarev, D. Schertzer, S. Lovejoy, and Y. Chigirinskaya. Unified multifractal atmospheric dynamics tested in the tropics: part ii, vertical scaling and generalized scale invariance. *Nonlinear Processes in Geophysics*, 1(2/3):115–123, 1994. doi: 10.5194/npg-1-115-1994. URL `https://www.nonlin-processes-geophys.net/1/115/1994/`.

S. Lovejoy and D. Schertzer. *The Weather and Climate: Emergent Laws and Multifractal Cascades*. Cambridge University Press, 2013.

B. B. Mandelbrot and C. J. G. Evertsz. *Exactly Self-similar Left-sided Multifractals*, pages 367–399. Springer Berlin Heidelberg, Berlin, Heidelberg, 1996. ISBN 978-3-642-84868-1.

A. Misiorek and R. Weron. *Heavy-Tailed Distributions in VaR Calculations*, pages 1025–1059. Springer Berlin Heidelberg, Berlin, Heidelberg, 2012.

S. B. Pope. *Turbulent flows*. Cambridge university press, 2000.

L. F. Richardson. *Weather Prediction by Numerical Process*. Cambridge University, Cambridge, 1922.

B. Ruddick, A. Anis, and K. Thompson. Maximum Likelihood Spectral Fitting: The Batchelor Spectrum. *Journal of Atmospheric and Oceanic Technology*, 17(11):1541–1555, 2000.

G. Samorodnitsky and M. S. Taqqu. Non-Gaussian Stable Processes: Stochastic Models with Infinite Variance. *Chapman and Hall, London*, 1994.

B. Scheifele, S. Waterman, L. Merckelbach, and J. R. Carpenter. Measuring the dissipation rate of turbulent kinetic energy in strongly stratified, low-energy environments: A case study from the arctic ocean. *Journal of Geophysical Research: Oceans*, 123(8):5459–5480, 2018. doi: 10.1029/2017JC013731. URL `https://agupubs.onlinelibrary.wiley.com/doi/abs/10.1029/2017JC013731`.

D. Schertzer, S. Lovejoy, and F. Schmitt. Structures in turbulence and multifractal universality. In *Small-Scale Structures in Three-Dimensional Hydrodynamic and Magnetohydrodynamic Turbulence*, pages 137–144. Springer, 1995.

F. G. Schmitt. Modeling of Turbulent Intermittency: Multifractal Stochastic Processes and Their Simulation. In L. Seuront and P. G. Strutton, editors, *Handbook of scaling methods in aquatic ecology: measurement, analysis, simulation*, chapter 29, pages 453–468. CRC Press, 2003.

W. D. Smyth, P. O. Zavialov, and J. N. Moum. Decay of turbulence in the upper ocean following sudden isolation from surface forcing. *Journal of Physical Oceanography*, 27(5):810–822, 1997.

H. E. Stanley and P. Meakin. Multifractal phenomena in physics and chemistry. *Nature*, 335(6189): 405, 1988.

A. F. Waterhouse, J. A. MacKinnon, J. D. Nash, M. H. Alford, E. Kunze, H. L. Simmons, K. L. Polzin, L. C. St. Laurent, O. M. Sun, R. Pinkel, L. D. Talley, C. B. Whalen, T. N. Huussen, G. S. Carter, I. Fer, S. Waterman, A. C. Naveira Garabato, T. B. Sanford, and C. M. Lee. Global patterns of diapycnal mixing from measurements of the turbulent dissipation rate. *Journal of Physical Oceanography*, 44(7):1854–1872, 2014. doi: 10.1175/JPO-D-13-0104.1. URL `https://doi.org/10.1175/JPO-D-13-0104.1`.

C. B. Whalen, J. A. MacKinnon, L. D. Talley, and A. F. Waterhouse. Estimating the mean diapycnal mixing using a finescale strain parameterization. *Journal of Physical Oceanography*, 45(4):1174–1188, 2015. doi: 10.1175/JPO-D-14-0167.1. URL `https://doi.org/10.1175/JPO-D-14-0167.1`.

A. M. Yaglom. The influence of fluctuations in energy dissipation on the shape of turbulent characteristics in the inertial interval. *Sov. Phys. Dokl.*, 11(26), 1966.